# Sex-specific pubertal and metabolic regulation of Kiss1 neurons via Nhlh2

**Silvia Leon[1,2†], Rajae Talbi[1,2†], Elizabeth A McCarthy[1,2], Kaitlin Ferrari[2], Chrysanthi Fergani[1,2], Lydie Naule[1,2], Ji Hae Choi[2], Rona S Carroll[1,2], Ursula B Kaiser[1,2], Carlos F Aylwin[3], Alejandro Lomniczi[3], Víctor M Navarro[1,2,4]***

[1]Harvard Medical School, Boston, United States; [2]Department of Medicine, Division of Endocrinology, Diabetes and Hypertension, Brigham and Women's Hospital, Boston, United States; [3]Division of Neuroscience, Oregon National Primate Research Center, Beaverton, United States; [4]Harvard Program in Neuroscience, Boston, United States

**Abstract** Hypothalamic Kiss1 neurons control gonadotropin-releasing hormone release through the secretion of kisspeptin. Kiss1 neurons serve as a nodal center that conveys essential regulatory cues for the attainment and maintenance of reproductive function. Despite this critical role, the mechanisms that control kisspeptin synthesis and release remain largely unknown. Using Drop-Seq data from the arcuate nucleus of adult mice and in situ hybridization, we identified Nescient Helix-Loop-Helix 2 (*Nhlh2*), a transcription factor of the basic helix-loop-helix family, to be enriched in Kiss1 neurons. JASPAR analysis revealed several binding sites for NHLH2 in the *Kiss1* and *Tac2* (neurokinin B) 5′ regulatory regions. In vitro luciferase assays evidenced a robust stimulatory action of NHLH2 on human *KISS1* and *TAC3* promoters. The recruitment of NHLH2 to the *KISS1* and *TAC3* promoters was further confirmed through chromatin immunoprecipitation. In vivo conditional ablation of *Nhlh2* from Kiss1 neurons using *Kiss1*^Cre:*Nhlh2*^fl/fl mice induced a male-specific delay in puberty onset, in line with a decrease in arcuate *Kiss1* expression. Females retained normal reproductive function albeit with irregular estrous cycles. Further analysis of male *Kiss1*^Cre:*Nhlh2*^fl/fl mice revealed higher susceptibility to metabolic challenges in the release of luteinizing hormone and impaired response to leptin. Overall, in Kiss1 neurons, Nhlh2 contributes to the metabolic regulation of kisspeptin and NKB synthesis and release, with implications for the timing of puberty onset and regulation of fertility in male mice.

*For correspondence:
vnavarro@bwh.harvard.edu

†These authors contributed equally to this work

**Competing interests:** The authors declare that no competing interests exist.

## Introduction

Hypothalamic Kiss1 neurons secrete kisspeptins (*Pinilla et al., 2012*), which act directly on gonado-tropin-releasing hormone (GnRH) neurons (*Irwig et al., 2004*) to stimulate GnRH release. GnRH then induces gonadotropin release from pituitary gonadotropes (*Herbison, 2006*). The absence of proper kisspeptin signaling leads to absent or delayed puberty, hypogonadism, and infertility (*de Roux et al., 2003*; *Seminara et al., 2003*). Kiss1 neurons are mainly found in two distinct hypothalamic nuclei, the arcuate nucleus (ARH) and the anteroventral periventricular nucleus (AVPV/PeN) (*Pinilla et al., 2012*). Kiss1^ARH neurons mediate the pulsatile release of GnRH (*Clarkson et al., 2017*; *Navarro et al., 2009*; *Wakabayashi et al., 2010*), while Kiss1^AVPV/PeN are almost exclusive to the female brain and control the surge-like release of GnRH (*Caraty et al., 2007*; *Sébert et al., 2010*).

Pulsatile GnRH release increases at the end of the juvenile period, determining the onset of puberty and the transition into adulthood (*Ojeda et al., 2010*). This process is largely dependent on the activation of Kiss1 neurons, in part, through the autosynaptic excitatory action of neurokinin B (NKB) (*Garcia et al., 2017*; *Gill et al., 2012*; *Navarro et al., 2012a*; *Ruiz-Pino et al., 2012*) and the inhibitory action of dynorphin through an autosynaptic alternation process that creates the

*kisspeptin pulse generator* (*Clarkson et al., 2017*; *Navarro et al., 2009*; *Plant, 2019*). However, the mechanisms that determine the initiation of the cascade of events that lead to activation of the kisspeptin pulse generator at the time of puberty onset, remain largely unknown.

Energy balance is a critical component in the control of puberty onset and maintenance of reproductive function in adulthood (*Navarro, 2020*). Deficient energy reserves are often associated with functional hypothalamic amenorrhea (*Laughlin et al., 1998*). A number of central and peripheral factors mediate the inhibition of the reproductive axis in conditions of negative energy balance. For example, the hunger signals agouti-related peptide (AgRP) and neuropeptide Y, produced in hypothalamic AgRP neurons, directly inhibit Kiss1 neurons under energetic deficit (*Padilla et al., 2017*). AgRP neurons, as well as Kiss1 neurons themselves, express the leptin receptor (Lepr) and are therefore direct targets of leptin, which is produced in the adipose tissue as a direct measure of the level of fat reserves (*Backholer et al., 2010*; *Cravo et al., 2013*; *Egan et al., 2017*; *Smith et al., 2006*; *True et al., 2011*). Deficient leptin signaling, as occurs in *ob/ob* or *db/db* mice, leads to obesity and infertility (*Donato et al., 2011b*; *Leshan et al., 2006*; *Navarro and Kaiser, 2013*). However, the central pathways that mediate the metabolic and reproductive roles of leptin remain largely unknown. Evidence from our lab and others indicate that the main control of both functions by leptin requires GABAergic neurons (*Martin et al., 2014*; *Vong et al., 2011*; *Zuure et al., 2013*), while the deletion of Lepr from glutamatergic neurons leads to a subtle metabolic and reproductive phenotype (*Martin et al., 2014*; *Vong et al., 2011*). Interestingly, Kiss1 neurons of the ARH are mostly glutamatergic (*Nestor et al., 2016*) and the deletion of *Lepr* from Kiss1 neurons does not prevent fertility in mice (*Cravo et al., 2013*; *Donato et al., 2011a*). These data suggest that leptin plays its primary reproductive role upstream of Kiss1 neurons and that the expression of Lepr in these neurons is limited to a modulatory role, likely in the acute regulation of kisspeptin release under specific metabolic conditions.

In addition to leptin, other peripheral factors that signal energetic state can directly act on Kiss1 neurons as well, such as insulin (*Qiu et al., 2015*), suggesting that Kiss1 neurons are a nodal target for metabolic cues (*Navarro, 2020*). Thus, the combined action of the metabolic signals that inform the brain of sufficient fuel reserves ultimately control the expression of the genes critical for kisspeptin release, that is, *Tac2* and *Pdyn* (which encode NKB and dynorphin A, respectively), and therefore the pulsatile release of kisspeptin (*Clarkson et al., 2017*; *Plant, 2019*; *Navarro, 2012b*). Conditions of negative energy balance rapidly and potently inhibit the reproductive axis, preventing or delaying puberty onset and fertility (*Navarro, 2020*; *Castellano et al., 2011*; *Luque et al., 2007*; *Roa et al., 2008*). Upon restoration of energy reserves, kisspeptin pulses resume and the reproductive axis becomes reactivated (*Navarro, 2020*). Still, despite the significant burden that metabolic imbalances impose on Kiss1 neurons, the intracellular mechanisms, for example, transcription factors (TFs), that translate this metabolic information have not yet been fully characterized. In vitro studies in cancer cell lines have identified SP-1 and AP-2α as activators of the *Kiss1* promoter (*Mitchell et al., 2006*; *Mitchell et al., 2007*), while in vivo only TTF1 and CUX1-p200 in the human (*Mueller et al., 2011*), and Tbx3 and Crtc1 in the mouse (*Sanz et al., 2015*; *Altarejos et al., 2008*), have been identified as potential enhancers of *Kiss1* transcription despite the large number of potential TF binding sites that have been described in the *Kiss1* promoter (*Goto et al., 2015*). Of these TFs, only Crtc1 (*Altarejos et al., 2008*) has been involved in the transcriptional regulation of *Kiss1* by metabolic cues, which suggests that most of the regulatory mechanisms at the *Kiss1* promoter level remain unknown.

In this study, through a series of in vitro and in vivo genetic and functional studies in male and female mice we identified the basic helix-loop-helix (bHLH) TF (*Jones, 2004*), Nhlh2, as a highly specific TF of Kiss1 neurons of the adult mouse brain that controls puberty onset in males, but not females, and conveys the information of the metabolic state to the reproductive axis.

## Results

### Nhlh2 is enriched in Kiss1 neurons of adult mice

Using a previously described database of ARH transcripts from 20,921 cells from adult wildtype (WT) male and female mice (*Campbell et al., 2017*), we identified the bHLH TF *Nhlh2*, as a highly enriched transcript in arcuate Kiss1 neurons. The enrichment is presented in a t-distributed

stochastic neighbor embedding (tSNE) plot comparing the expression of a highly specific marker of Kiss1[ARH] neurons (*Tac2*) with *Nhlh2* (*Figure 1—figure supplement 1A*). As reported previously, *Nhlh2* is also present in a fraction of proopiomelanocortin (POMC) neurons (*Schmid et al., 2013*; *Figure 1—figure supplement 1A*). In situ hybridization, though RNAscope assay of *Kiss1* and *Nhlh2* confirmed this co-expression in 71% of Kiss1[ARH] neurons of adult WT males (*Figure 1A,D*), 39% of Kiss1[ARH] neurons of adult ovariectomized (OVX) WT females (*Figure 1B,D*) and 16% of Kiss1[AVPV/PeN] neurons of OVX+E2 replaced WT females (*Figure 1C,D*). These data are in line with the expression assay of *Nhlh2* in adult male mice performed by the Allen Brain Atlas, which shows the concentrated expression of *Nhlh2* to the ARH (*Figure 1—figure supplement 1B*).

## Nhlh2 is recruited to the *KISS1* and *TAC3* promoters and enhances their activity

Kiss1[ARH] neurons express NKB in addition to kisspeptin (*Lehman et al., 2010*). In order to assess the potential regulatory role of NHLH2 on *Kiss1* and *Tac2/3* function, we performed JASPAR analysis (*Fornes et al., 2020*; *Gearing et al., 2019*) on the human, mouse and rat 5′ regulatory regions of these genes, using two available binding matrices (*Figure 2A*). Our analysis revealed that there are several putative NHLH2 binding sites in the *Kiss1* and *Tac2/3* genes in all species studied. Moreover, the *Tac2/3* genes seemed more enriched at an average of 1 site every 80 nt while the *Kiss1* promoter region contained 1 site every 150 nt (*Figure 2B*).

The fact that *Nhlh2* is expressed in Kiss1 neurons and the identification of binding sites in *Kiss1* and *Tac2* promoters led us to hypothesize that NHLH2 could regulate their expression. To test this hypothesis, we performed promoter assays with human *KISS1* or *TAC3* (equivalent to *Tac2* in rodents) promoters and increasing amounts of human NHLH2. Transfection of either promoter resulted in increased luciferase activity compared with the empty pGL2 vector, this activity was

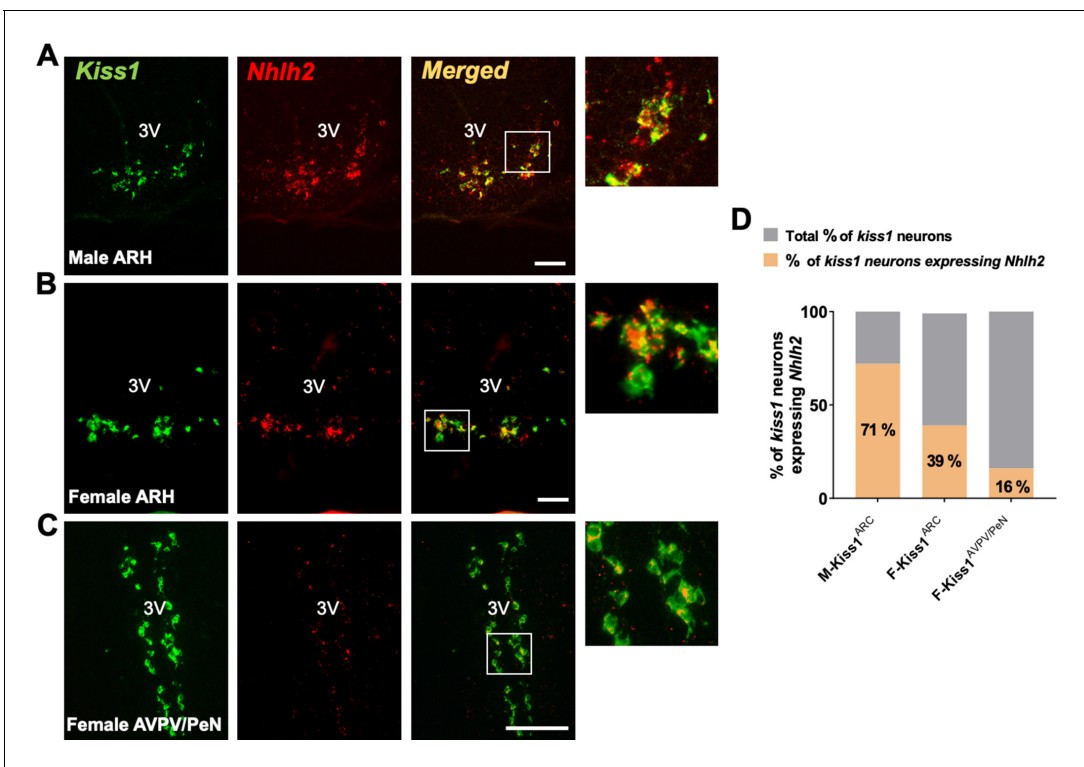

**Figure 1.** Nhlh2 is a marker of Kiss1 neurons. (A) In situ hybridization (RNAscope) of *Kiss1* and *Nhlh2* in the ARH of adult male (A) and ovariectomized female mice (B), and in the AVPV/PeN of ovariectomized+estradiol treated female mice (C). (D) Percentage of Kiss1 neurons expressing *Nhlh2*. Scale bar=100 μm.

The online version of this article includes the following figure supplement(s) for figure 1:

**Figure supplement 1.** *Nhlh2* is enriched in Kiss1 neurons of adult mice.

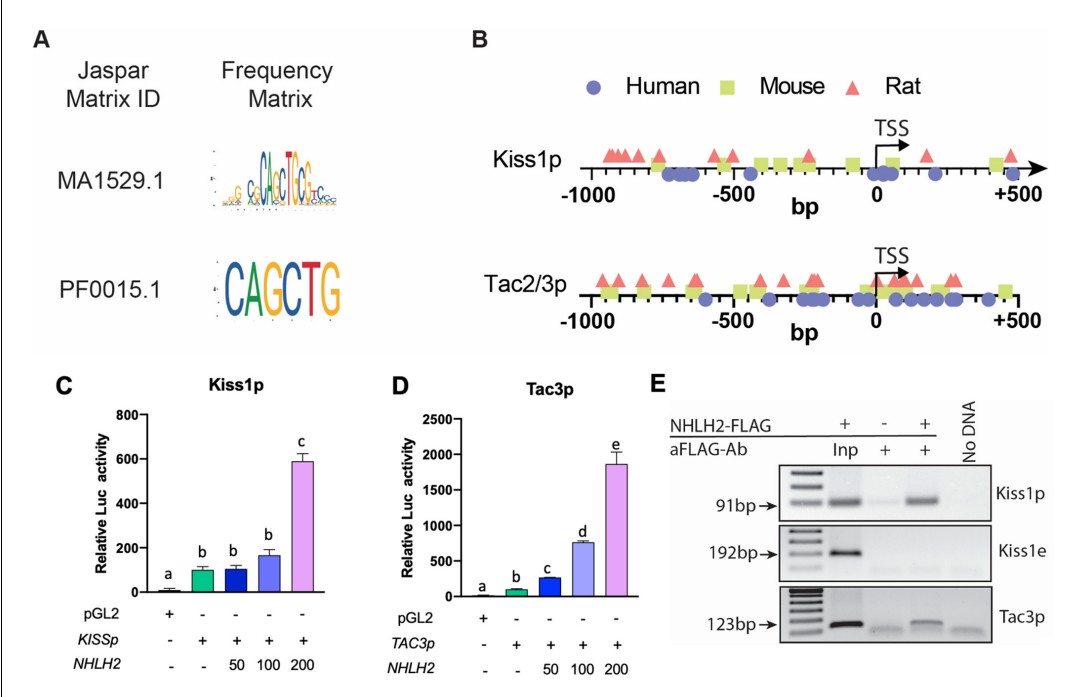

**Figure 2.** NHLH2 is recruited to and stimulates *KISS1* and *TAC3* promoter activity. (**A, B**) Identification of NHLH2 biding sites in *Kiss1* and *Tac2/3*. NHLH2 DNA recognition sites as identified in two Jaspar matrices (**A**). Distribution of NHLH2 binding sites in human, mouse, and rat Kiss1 promoters (Kiss1p) and TAC2/3 promoters (Tac2/3 p) (**B**). (**C–E**) Effect of NHLH2 on luciferase activity regulated by the human *KISS1* (**C**) and *TAC3* (**D**), promoter (p). Neuro-2a cells were transfected with luciferase reporter constructs containing the 5′ flanking region of the indicated genes. After 48 h, the cells were harvested and assayed for luciferase activity. Bars represent mean± SEM (n=three biological replicates per group). Groups with different letters are significantly different (P<0.05), as determined by one-way ANOVA followed by the Student-Newman-Keuls test. (**E**) Association of NHLH2 with the 5′ promoter region of the *KISS1* (Kiss1p) and *TAC3* (Tacr3p) genes as determined by PCR amplification of DNA immunoprecipitated with antibodies recognizing the FLAG/OctA epitope tagging NHLH2. As negative control, a region of the *Kiss1* promoter outside of the area containing Nhlh2 E-box was assayed (Kiss1e). Inp=Input. Data presented as the mean± SEM.

The online version of this article includes the following source data for figure 2:

**Source data 1.** Association of NHLH2 with the 5′ promoter regeion of KISS1.

**Source data 2.** Association of NHLH2 with a region of the Kiss1 promoter outside of the area containing Nhlh2 E-box was assayed (Kiss1e).

**Source data 3.** Association of NHLH2 with the 5′ promoter regeion of TAC3.

**Source data 4.** Images of uncropped geles.

significantly enhanced by NHLH2 co-expression, reaching a 6-fold and a 20-fold increase in *KISS1* and *TAC3* promoter activity, respectively, at the highest NHLH2 concentration (*Figure 2C,D*). Interestingly, the *TAC3* promoter was more sensitive to NHLH2 stimulation, since the promoter was activated even at the lowest doses (*Figure 2D*), in line with the JASPAR analysis. Furthermore, chromatin immunoprecipitation (ChIP) assays showed that NHLH2 directly interacts with the *Kiss1* and *Tac3* immediate 5′-regulatory regions, as opposed to an enhancer site located ~3 kb upstream of the *Kiss1* TSS, where no binding sites were detected (*Figure 2E*).

## *Nhlh2* expression decreases in the ARH during postnatal development

*Nhlh2* is expressed throughout the hypothalamus during the early developmental period (*Cogliati et al., 2007*). However, Drop-seq and in situ hybridization data of adult animals showed a highly specific expression of *Nhlh2* to the ARH and mainly to Kiss1 neurons. Thus, we hypothesized that the overall expression of *Nhlh2* throughout development would decrease prior to puberty onset as it becomes specific to some ARH subpopulations, that is, Kiss1 neurons and a fraction of POMC neurons, in adulthood. To assess the expression profile of *Nhlh2* postnatally, mediobasal hypothalamic (MBH) samples—where the ARH is located—from WT mice were assessed at different

postnatal time points: infantile, early juvenile, late juvenile, and prepubertal. *Kiss1* expression in the MBH of male (*Figure 3A*) and female (*Figure 3C*) mice did not experience any significant change in expression, as previously described (*Gill et al., 2012*). *Nhlh2* expression decreased significantly from early juvenile onwards in males and late juvenile in females (*Figure 3B and D*), suggesting a possible downregulation of the expression by increasing circulating levels of sex steroids.

### Absence of *Nhlh2* from Kiss1 neurons disrupts estrous cycles in females and delays puberty onset in male mice

To fully characterize the role of Kiss1-neuron Nhlh2 in the control of reproduction, we generated a Kiss1-specific Nhlh2 knockout mouse (*Kiss1^Cre^:Nhlh2^fl/fl^*). The absence of Nhlh2 from Kiss1 neurons, as well as the absence of global recombination, was confirmed through in situ hybridization

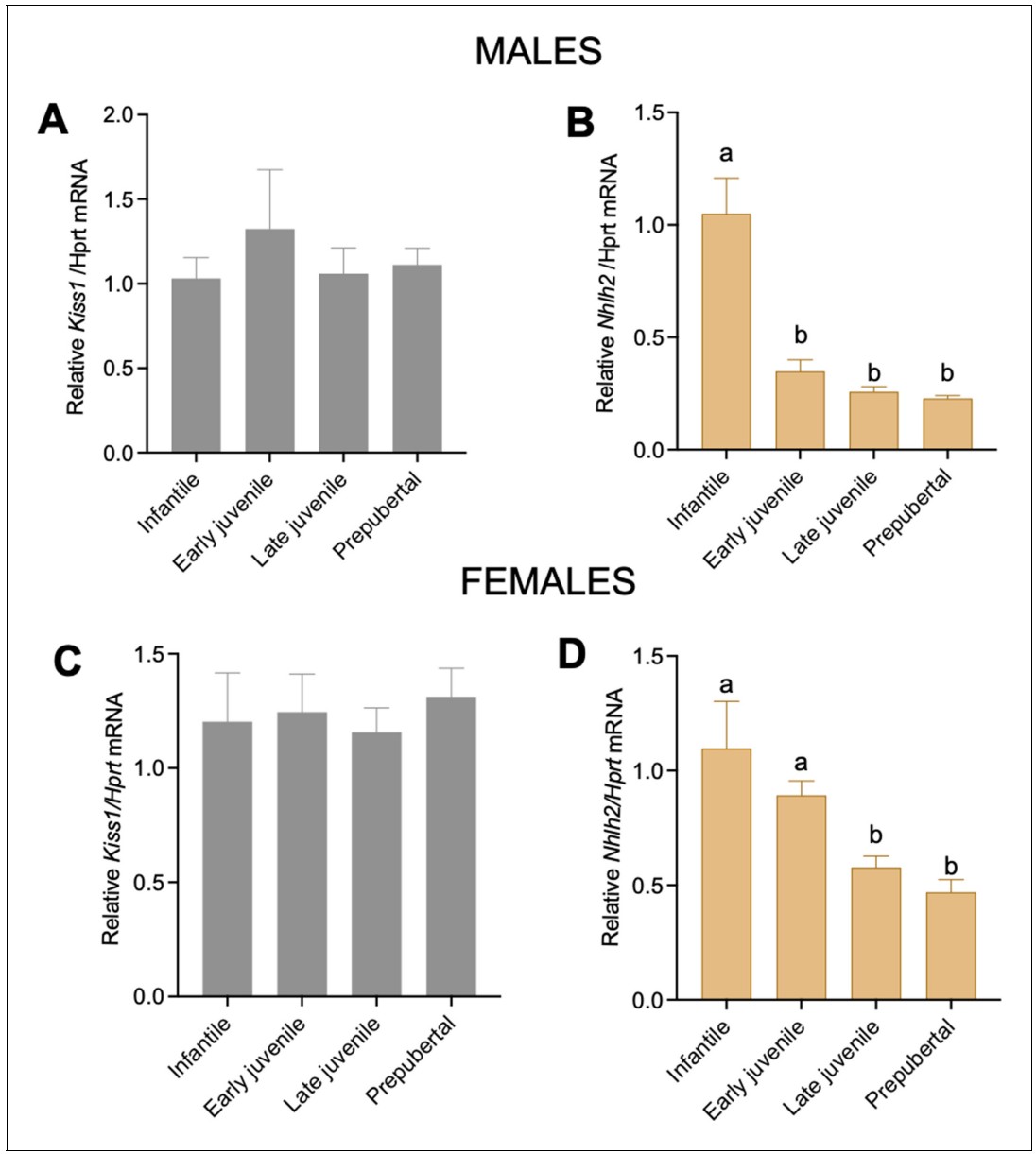

**Figure 3.** Gene expression of *Kiss1* and *Nhlh2* in the ARH during the prepubertal period. Expression profile of *Kiss1* and *Nhlh2* in the MBH of WT male (n=5/group) (A, B) and female (n=6/group) (C, D) mice at different postnatal time points: infantile, early juvenile, late juvenile, and prepubertal, normalized to the housekeeping gene *Hprt*. Groups with different letters are significantly different ($p<0.05$), as determined by one-way ANOVA followed by the Student-Newman-Keuls test. Data presented as the mean± SEM.

(RNAscope) (*Figure 4—figure supplement 1A*), which showed that *Kiss1^Cre^:Nhlh2^fl/fl^* mice still display detectable levels of *Nhlh2* in other arcuate neurons, likely the fraction of POMC neurons that express Nhlh2 (*Fox et al., 2007*; *Vella et al., 2007*). Importantly, the expression levels of *Nhlh2* in the gonads of *Kiss1^Cre^:Nhlh2^fl/fl^* mice were similar to the control group (*Figure 4—figure supplement 1B and C*), supporting the specific deletion of *Nhlh2* in Kiss1 neurons only.

Female *Kiss1^Cre^:Nhlh2^fl/fl^* mice displayed normal timing of puberty onset, determined through the day of vaginal opening (VO) and first estrus (*Figure 4A–D*). Further examination of the estrous cycles for 30 days after VO revealed a significant disruption in estrous cyclicity, with *Kiss1^Cre^:Nhlh2^fl/fl^* mice spending more time in diestrus and less time in estrus (*Figure 4E,F*). Because females have a

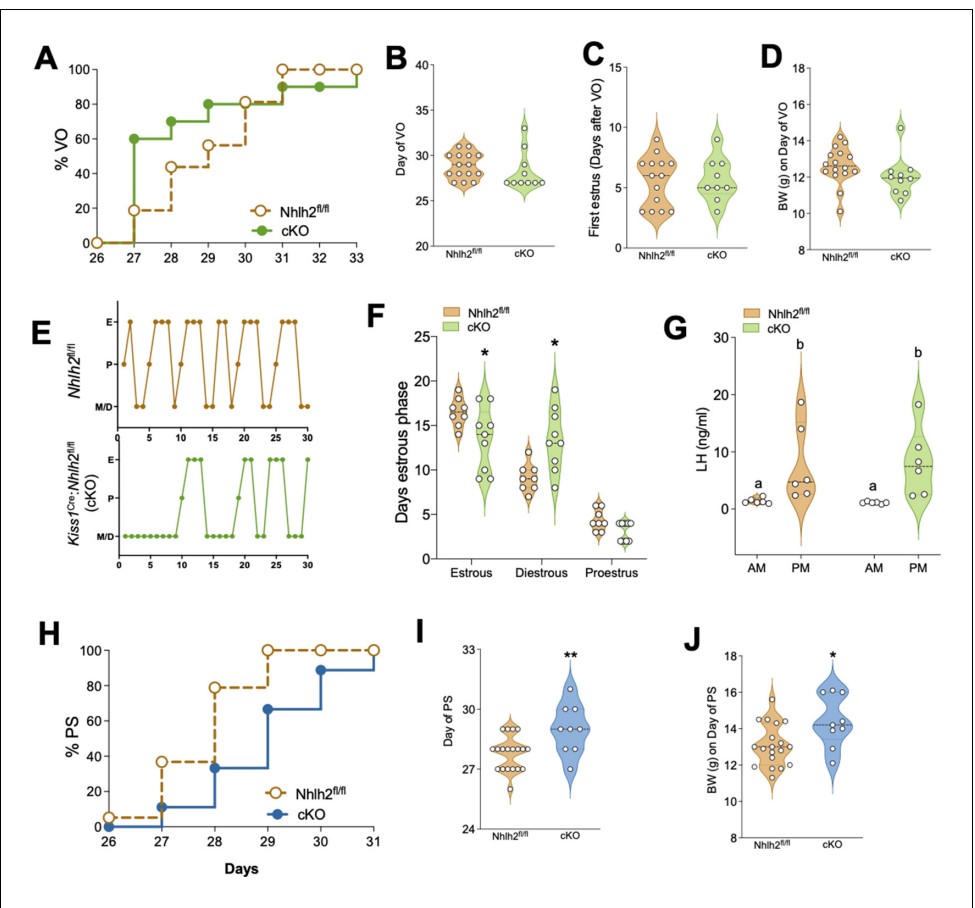

**Figure 4.** *Kiss1^Cre^:Nhlh2^fl/fl^* male mice present delayed puberty onset while females display irregular estrous cycles. Female *Kiss1^Cre^:Nhlh2^fl/fl^* mice present normal puberty onset as assessed by daily monitoring of vaginal opening (VO) and documented by cumulative percent of animals at VO (**A**), mean age of VO (**B**) *Nhlh2^fl/fl^* (n=15) and *Kiss1^Cre^:Nhlh2^fl/fl^* (n=10) and first estrus (**C**) *Nhlh2^fl/fl^* (n=14) and *Kiss1^Cre^:Nhlh2^fl/fl^* (n=9). These females have normal body weight (BW) at the age of VO (**D**). Estrous cyclicity, assessed by daily vaginal cytology for 30 days is irregular in *Kiss1^Cre^:Nhlh2^fl/fl^* mice (**E, F**), presenting longer time in diestrus compared to control, *Nhlh2^fl/fl^* (n=8) and *Kiss1^Cre^:Nhlh2^fl/fl^* (n=9). *p<0.05 two-way ANOVA followed by Bonferroni test. (**G**) LH measurements in the morning (AM [10 a.m.]) and afternoon (PM [7:30 p.m.]) after lights off of female mice subjected to an LH surge induction protocol. n=6/group. Groups with different letters are significantly different. Two-way ANOVA followed by Bonferroni post hoc test. (**H, I**) Pubertal development of male *Kiss1^Cre^:Nhlh2^fl/fl^* mice and control littermates determined by preputial separation. *p<0.05 by Student's t-test. (**J**) BW of males at the age of preputial separation. *Nhlh2^fl/fl^* (n=19) and *Kiss1^Cre^:Nhlh2^fl/fl^* (n=9). *p<0.05 by Student's t-test. Data presented as median (dotted line)+distribution of the data and its probability density (violin plot). cKO=*Kiss1^Cre^:Nhlh2^fl/fl^*. LH, luteinizing hormone.

The online version of this article includes the following figure supplement(s) for figure 4:

**Figure supplement 1.** Validation of the *Kiss1^Cre^:Nhlh2^fl/fl^* mouse model.

**Figure supplement 2.** *Kiss1^Cre^:Nhlh2^fl/fl^* mice display normal body weight (BW).

larger population of Kiss1 neurons in the AVPV/PeN than males, which mediates the generation of the luteinizing hormone (LH) surge (*Smith et al., 2005*), and a fraction of them expresses *Nhlh2*, *Figure 1*, we assessed whether the absence of Nhlh2 from Kiss1[AVPV/PeN] neurons (and from Kiss1[ARH] neurons) affects the generation of the LH surge. *Kiss1[Cre]:Nhlh2[fl/fl]* and control (*Nhlh2[fl/fl]*) mice were submitted to an LH induction protocol. This experiment revealed a similar magnitude of the LH surge in both groups (*Figure 4G*), indicating that Nhlh2 in Kiss1 neurons is not necessary for the generation of the preovulatory LH surge in mice.

In males, the specific deletion of *Nhlh2* from Kiss1 neurons led to a significant delay in puberty onset, as observed by preputial separation (*Figure 4H,I*). Due to the older age, the body weight (BW) at the time of puberty onset in *Kiss1[Cre]:Nhlh2[fl/fl]* males was slightly higher than in controls (*Figure 4J*). However, the overall BW throughout development did not change between genotypes in either sex during 130 days of study postnatally under regular chow diet (*Figure 4—figure supplement 2A and B*). Because Kiss1 neurons have been involved in the control of energy balance (*Navarro, 2020*; *Tolson et al., 2014*), we exposed these mice to a 60% high-fat diet (HFD) for 11 additional weeks. This metabolic challenge did not reveal any significant difference in BW between groups (*Figure 4—figure supplement 2A and B*).

## *Kiss1[Cre]:Nhlh2[fl/fl]* mice are fertile

To determine the impact of Nhlh2 removal from Kiss1 neurons on fertility under normal chow diet, male and female *Kiss1[Cre]:Nhlh2[fl/fl]* mice were mated with sexually experienced WT mates for 3 months. *Kiss1[Cre]:Nhlh2[fl/fl]* mice retained normal fertility compared to the control as evidenced by their parturition latency, number of pups per litter, and number of litters in 3 months (*Figure 5A–C*). However, male *Kiss1[Cre]:Nhlh2[fl/fl]* mice presented a variable phenotype with a fraction of them showing absent or severely reduced fertility (*Figure 5A–C*). These data are in line with the normal gonadal histology in *Kiss1[Cre]:Nhlh2[fl/fl]* mice. Testes of *Kiss1[Cre]:Nhlh2[fl/fl]* males show mature sperm, and ovaries of *Kiss1[Cre]:Nhlh2[fl/fl]* females presented similar numbers of growing follicles and corpora lutea (CL) as controls (*Figure 5—figure supplement 1*).

## Expression of KNDy genes in *Kiss1[Cre]:Nhlh2[fl/fl]* mice

Expression analysis of the KNDy genes revealed a significant decrease in the expression of *Kiss1* in the MBH of male and female *Kiss1[Cre]:Nhlh2[fl/fl]*, which was further observed by RNAscope (*Figure 6A–D*). Interestingly, while the expression of the rest of the KNDy genes (*Tac2* and *Pdyn*) remained similar to controls in *Kiss1[Cre]:Nhlh2[fl/fl]* males (*Figure 6A*), there was a significant decrease

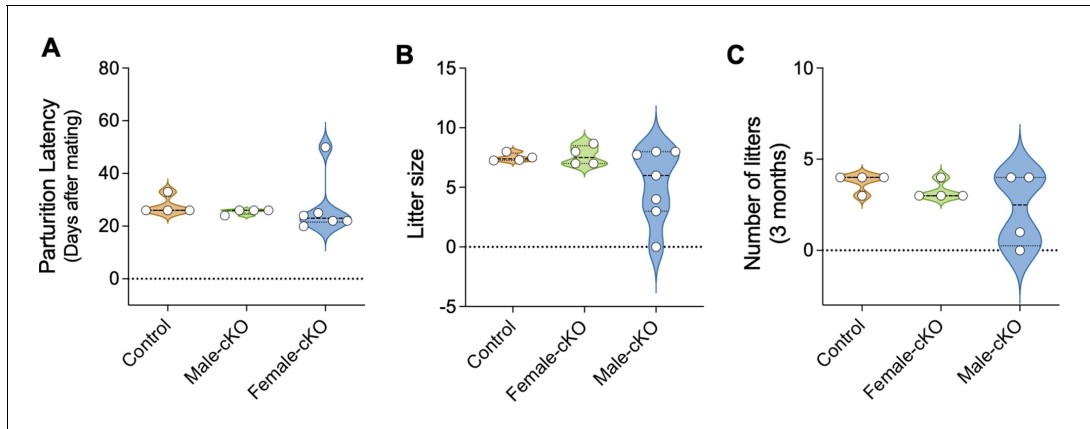

**Figure 5.** *Kiss1[Cre]:Nhlh2[fl/fl]* mice are fertile. Male and female *Kiss1[Cre]:Nhlh2[fl/fl]* mice were mated with sexually experienced WT mates for 3 months. The time to deliver pups, that is, parturition latency (**A**), total number of pups in 3 months (**B**), and number of litters in 3 months (**C**) were monitored, n=4–7/ group. Two-way ANOVA followed by Bonferroni post hoc test. Values are presented as median (middle line)±max/min (violin plot). cKO=*Kiss1[Cre]: Nhlh2[fl/fl]*. WT, wild-type.

The online version of this article includes the following figure supplement(s) for figure 5:

**Figure supplement 1.** Histological analysis of testes and ovaries from *Kiss1[Cre]:Nhlh2[fl/fl]* mice.

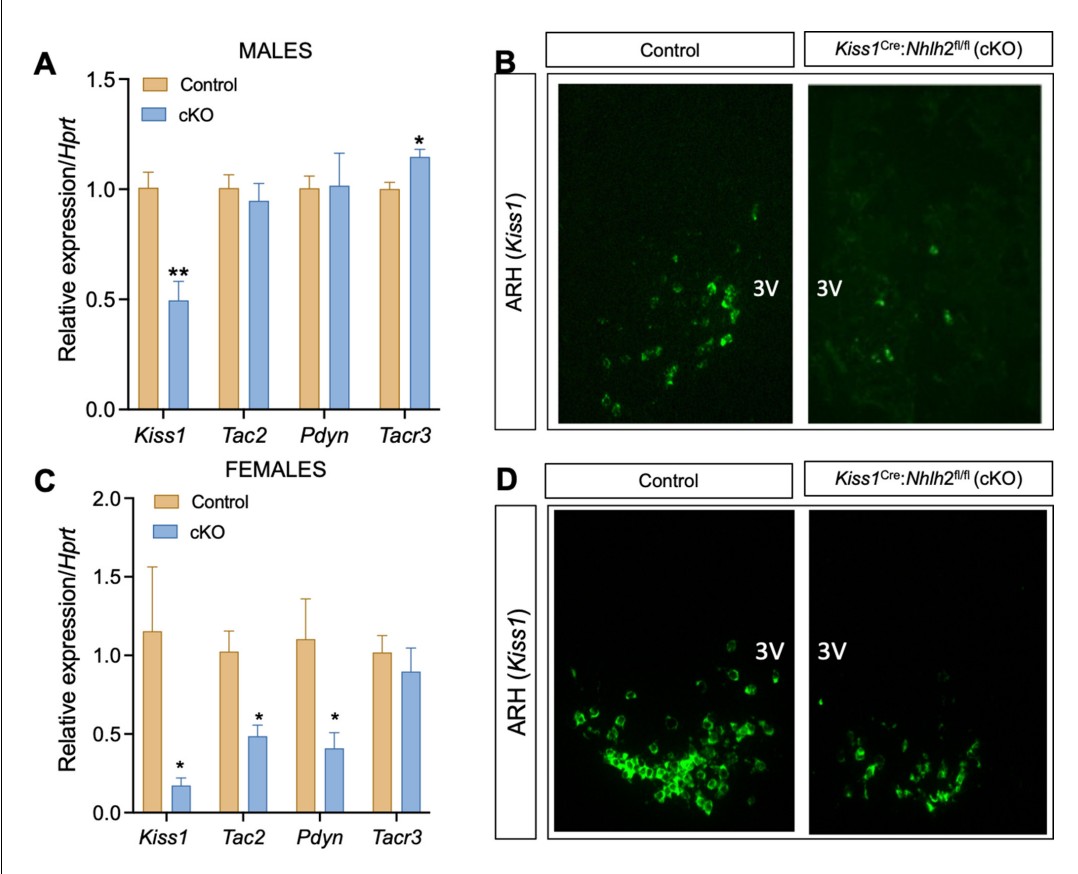

**Figure 6.** Expression of KNDy genes in the ARH of *Kiss1^{Cre}:Nhlh2^{fl/fl}* mice. The expression of the KNDy genes (*Kiss1, Tac2,* and *Pdyn*) and the receptor for NKB (*Tacr3*) were assessed in the MBH of adult male (**A**) and female (**C**) *Kiss1^{Cre}:Nhlh2^{fl/fl}* mice and their control (Kiss1^{Cre/+}) littermates (n=4/group; *p<0.05, **p<0.01. Two-way ANOVA followed by Bonferroni test). Representative images of in situ hybridization (RNAscope) depicting the expression of *Kiss1* expression in the MBH of *Kiss1^{Cre}:Nhlh2^{fl/fl}* male (**B**) and female (**D**) mice compared to controls. Data presented as the mean± SEM. cKO=*Kiss1^{Cre}:Nhlh2^{fl/fl}*.

in the expression of both genes in females (*Figure 6C*). The NKB receptor (*Tacr3*) showed a modest but significant increase in the expression in *Kiss1^{Cre}:Nhlh2^{fl/fl}* males, while it remained unchanged in females.

## Impaired response to leptin in the absence of Nhlh2 in Kiss1 neurons

Despite the decrease in *Kiss1*, the central administration of kisspeptin-10 (Kp-10; 50 pmol) or the NK3R agonist senktide (600 pmol) in males induced similar responses in terms of LH release in both genotypes (*Figure 7A,B*). Moreover, the response of males to gonadectomy (GDX) after 10 days was also similar between both groups (*Figure 7C*). These data suggest that at the doses used of Kp-10 and senktide, male *Kiss1^{Cre}:Nhlh2^{fl/fl}* mice are able to release enough kisspeptin and GnRH to induce gonadotropin release, in line with the overcoming of their delayed puberty and eventual fertility observed in males (*Figures 4* and *5*).

Because Nhlh2 has been described to interact with STAT3, the second messenger pathway induced by the activation of Lepr (*AL_Rayyan et al., 2014*), we tested the response of *Kiss1^{Cre}: Nhlh2^{fl/fl}* male mice to two metabolic challenges. First, mice were castrated for a week to increase gonadotropin release and then subjected to a 24 hr fasting protocol (in order to uncover the inhibitory mechanisms of negative energy balance on LH release), followed by 12 hr of refeeding. This protocol led to a significantly faster reduction in circulating LH levels in *Kiss1^{Cre}:Nhlh2^{fl/fl}* mice than in *Nhlh2^{fl/fl}* controls (*Figure 7D*), suggesting that these mice are more sensitive to decreasing levels of energy reserves. However, the response to refeeding was similar in both groups, indicating that metabolic cues can still activate Kiss1 neurons in an Nhlh2-independent manner within Kiss1 neurons

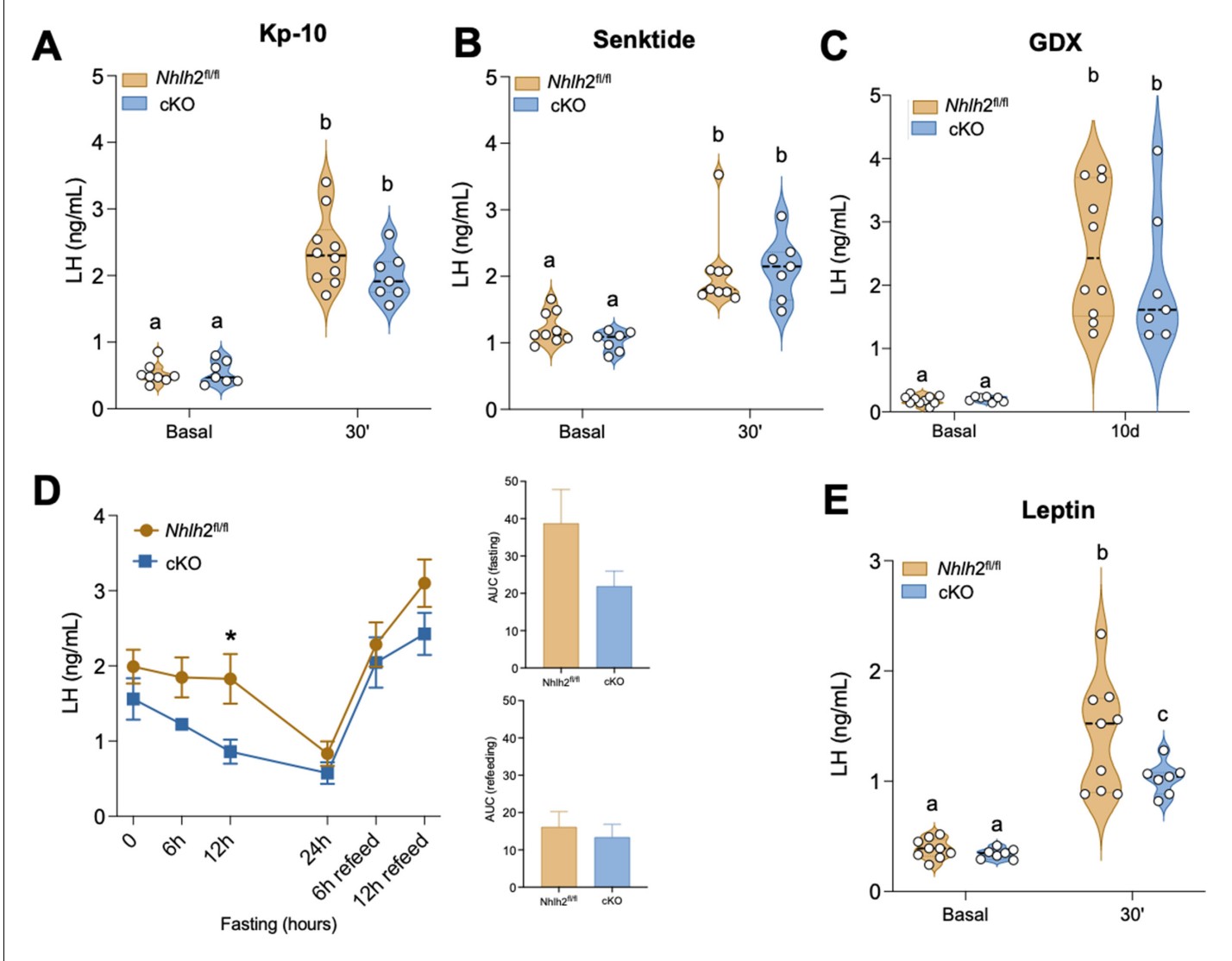

**Figure 7.** Functional assessment of Kiss1 neurons in *Kiss1^Cre^:Nhlh2^fl/fl^* male mice under fed and fasting conditions. Central (icv) administration of Kp-10 (50 pmol/mouse in 5 µl) (*Nhlh2^fl/fl^* [n=10] and *Kiss1^Cre^:Nhlh2^fl/fl^* [n=7]) (**A**) or senktide (600 pmol/mouse in 5 µl) Nhlh2^fl/fl^ (n=9) and *Kiss1^Cre^:Nhlh2^fl/fl^* (n=7) (**B**) was performed in adult male *Kiss1^Cre^:Nhlh2^fl/fl^* mice and controls. Blood samples were collected before and 30 min after treatment. Groups with different letters are significantly different two-way ANOVA followed by Bonferroni test. In (**C**), the ability of *Kiss1^Cre^:Nhlh2^fl/fl^* male mice to generate a compensatory LH rise after gonadectomy was assessed before surgery and 10 days post-gonadectomy. *Nhlh2^fl/fl^* (n=10) and *Kiss1^Cre^:Nhlh2^fl/fl^* (n=7). Two-way ANOVA followed by Bonferroni test. (**D**) Response of LH in gonadectomized male mice during 24 hr of fasting and 12 hr of refeeding (*Nhlh2^fl/fl^* [n=10] and *Kiss1^Cre^:Nhlh2^fl/fl^* [n=7]); *p<0.05 compared with the control group at 12 hr. Two-way ANOVA followed by Fisher's test. The area under the curve (AUC) was determined during fasting and refeeding periods. (**E**) Response of mice to overnight fasting followed by leptin administration. Blood samples were collected before and 30 min after leptin (2 µg/mouse in 5 µl) treatment. *Nhlh2^fl/fl^* (n=9) and *Kiss1^Cre^:Nhlh2^fl/fl^* (n=6); Groups with different letters are significantly different. Two-way ANOVA followed by Bonferroni post hoc test. Data presented as median (dotted line)+distribution of the data and its probability density (violin plot) or the mean± SEM (in (**D**)). cKO=*Kiss1^Cre^:Nhlh2^fl/fl^*. LH, luteinizing hormone.

or indirectly through other neurons, for example, POMC. Thus, in order to better assess if the response of Kiss1 neurons to leptin is impaired in *Kiss1^Cre^:Nhlh2^fl/fl^* mice—as indicated by studies describing Nhlh2/STAT3 interaction (*Vella et al., 2007*), we administered leptin after overnight fasting and collected serum LH samples a short time after (30 min). The results showed that *Kiss1^Cre^: Nhlh2^fl/fl^* mice displayed a significantly lower response to leptin compared to controls (*Figure 7E*), thus supporting the contention that Nhlh2 serves as a mechanism for Kiss1 neurons to respond to leptin action, likely through its interaction with STAT3 (*AL_Rayyan et al., 2014*).

## Discussion

Through Drop-Seq single-cell transcriptome analysis of adult mouse arcuate samples (*Campbell et al., 2017*), we have identified the bHLH (*Jones, 2004*) TF Nhlh2 as a marker of Kiss1[ARH] neurons in the adult mouse brain. Neuronal types were clustered depending on the expression of thousands of genes that varied the most in the data set. Among those, *Nhlh2* appeared as the gene that strongly marked Kiss1 neurons, being the most enriched transcript in these neurons ahead of the KNDy co-transmitters *Tac2* and *Pdyn* (*Lehman et al., 2010*). Nhlh2 is highly expressed in Kiss1 neurons from the time of their initial development during the embryonic phase (*Huisman et al., 2019*), and our data indicate that this co-expression is retained into adulthood. Moreover, we demonstrate that Nhlh2 potently enhances the activity of *KISS1* and *TAC3* promoters, suggesting a potential role in the central activation of the reproductive axis by controlling kisspeptin output.

Nhlh2 has been shown to be involved in the development of the brain, including GnRH and POMC neurons (*Schmid et al., 2013*; *Cogliati et al., 2007*; *Fox et al., 2007*; *Vella et al., 2007*; *Good et al., 1997*; *Johnson et al., 2004*; *Krüger et al., 2004*). *Nhlh2* KO male mice are infertile, presenting small penises at weaning, cryptorchidism, and no signs of puberty (*Good et al., 1997*). In the same vein, *Nhlh2* KO females are hypogonadal, displaying small ovaries, thread-like uteri, delayed puberty onset (first estrus), and irregular cycles when reared alone, but they are fertile when reared in the presence of males (*Cogliati et al., 2007*; *Good et al., 1997*; *Johnson et al., 2004*). Interestingly, while Nhlh2 has been implicated in the migratory process of a fraction of GnRH neurons (*Cogliati et al., 2007*; *Johnson et al., 2004*), *Nhlh2* KO mice still show large amounts of GnRH neurons in the preoptic area (well above the 12% of GnRH neurons that is required for normal reproductive function; *Herbison et al., 2008*). Moreover, specific ablation of *Nhlh2* from GnRH neurons using *Gnrh1*[Cre]:*Nhlh2*[fl/fl] mice displayed normal reproductive function despite a 20% reduction in the number of GnRH neurons (*Schmid et al., 2013*). In addition, GnRH neurons do not express *Nhlh2* beyond embryonic day 18.5E (*Cogliati et al., 2007*), supporting a likely role in the development (migration) of a small fraction of GnRH neurons but not in their function during adulthood. These data, along with the absence of *Nhlh2* in most of the adult mouse brain, with the exception of the ARH, and within this nucleus, mostly in Kiss1 neurons (*Cogliati et al., 2007*; and *present data*), suggests a likely developmental and functional role of Nhlh2 in Kiss1 neurons that may contribute to the hypogonadal phenotype observed in *Nhlh2* KO mice. The developmental restriction of the expression of Nhlh2 in the hypothalamus is further evidenced by the decline in expression in the ARH from the infantile to prepubertal age, likely as a consequence of the regulation of circulating sex steroid levels on Nhlh2 expression (*Good and Braun, 2013*). It is plausible that Nhlh2 be a part of the machinery that regulates KNDy neurons during the negative feedback of sex steroids, although this hypothesis remains to be investigated.

In line with the data observed in whole-body *Nhlh2* KO (*Johnson et al., 2004*), *Kiss1*[Cre]:*Nhlh2*[fl/fl] mice showed a sexual dimorphism in their reproductive phenotype, which was more severe in males than in females. Males presented delayed puberty onset and different degrees of subfertility. This indicates that the role of Nhlh2 within Kiss1 neurons might be frequently compensated by other, yet unknown, TFs that lead to the gaining of reproductive function in females and in the majority of males. It has been demonstrated that 5% of kisspeptin production is sufficient to maintain reproductive capabilities (*Popa et al., 2013*), which may explain the overall fertile phenotype of *Kiss1*[Cre]:*Nhlh2*[fl/fl] mice despite the reduction in *Kiss1* expression observed in *Kiss1*[Cre]:*Nhlh2*[fl/fl] mice. *Kiss1*[Cre]:*Nhlh2*[fl/fl] females displayed normal timing of puberty onset and fertility although their estrous cycles were disrupted under normal chow diet, resembling the phenotype of whole-body *Nhlh2* KO females reared with males (*Good et al., 1997*). Of note, females in our study were reared with their male counterparts until weaning, which may have contributed to their normal reproductive phenotype as suggested in these earlier studies (*Good et al., 1997*). Whether *Kiss1*[Cre]:*Nhlh2*[fl/fl] females present reduced reproductive longevity, as also documented for whole-body *Nhlh2* KO females (*Johnson et al., 2004*), was not assessed in this study.

The origin of the sexual dimorphism in the impact of Nhlh2 action in Kiss1 neurons is unclear. All mice respond normally to the stimulation of Kiss1 neurons after a senktide challenge and the post-gonadectomy raise of LH is preserved, indicating sufficient output of kisspeptin to elicit full gonadotropin responses under normal fed ad libitum conditions. However, the hypothalamic Kiss1 system is

sexually differentiated, and the population of Kiss1 neurons in the preoptic area (Kiss1$^{AVPV/PeN}$), which is virtually absent in males (*Smith et al., 2005*), minimally expresses Nhlh2 in females. Thus, the maintenance of an intact population of Kiss1$^{AVPV/PeN}$ neurons in females may account for the preservation of the timing of puberty onset and fertility—as further evidenced by their ability to mount a normal preovulatory LH surge.

Previous studies have also described metabolic impairments (obesity) in *Nhlh2* KO mice due to the decrease in the number of POMC neurons (*Vella et al., 2007*). Interestingly, the percentage of co-expression of *Pomc* and *Nhlh2* in adulthood is low (*Fox et al., 2007*; *Vella et al., 2007*; *Good et al., 1997*), suggesting that this metabolic phenotype may be a developmental process that leads to the loss of POMC neurons rather than an effect during adulthood. The removal of Nhlh2 from Kiss1 neurons did not induce any metabolic phenotype in *Kiss1$^{Cre}$:Nhlh2$^{fl/fl}$* mice under regular or HFD chow, which supports a role for Nhlh2 in POMC neurons in the obesity phenotype of whole-body KOs. Interestingly, Kiss1 neurons have been implicated in the control of energy balance and BW (*Tolson et al., 2014*). However, the present data suggest that Nhlh2 is not involved in any potential metabolic role of Kiss1 neurons.

Nhlh2 interacts with STAT3 in the downstream mechanism of leptin's action (*AL_Rayyan et al., 2014*). Thus, we hypothesized that if Nhlh2 mediates the action of leptin to control the expression of *Kiss1* (and *Tac2*) directly in Kiss1$^{ARH}$ neurons, *Kiss1$^{Cre}$:Nhlh2$^{fl/fl}$* mice would be more sensitive than controls to decrease circulating leptin levels, which occurs quickly during a fasting paradigm (*Luque et al., 2007*). This was indeed the case during a 24 hr fasting protocol in mice with high circulating LH levels (gonadectomized) and suggests that Nhlh2 is part of the intracellular machinery of Kiss1 neurons that serves as a rapid sensor of circulating metabolites—at least for leptin. Refeeding increased circulating LH levels in both mouse groups, that is, controls and *Kiss1$^{Cre}$:Nhlh2$^{fl/fl}$* mice, equally within 6 hr. This increase in LH release depends on the increase in kisspeptin and GnRH release in *Kiss1$^{Cre}$:Nhlh2$^{fl/fl}$* mice, and therefore highlights the multi-level action of metabolic factors acting on different hypothalamic nuclei (*Lowell, 2019*), which eventually activate Kiss1 neurons quickly after the restoration of energetic resources. The exogenous administration of leptin after overnight fasting, without the influence of other metabolic cues, for example, insulin, uncovered an impaired response in LH release of *Kiss1$^{Cre}$:Nhlh2$^{fl/fl}$* mice to leptin. This finding supports the potential direct action of leptin on Kiss1 neurons and, therefore, the likely participation of Nhlh2 in the second messenger cascade of Lepr to activate kisspeptin (and potentially NKB) expression and release.

These data further contribute to the elucidation of the complex network of leptin action to control reproductive function and uncovers a potential effect of leptin directly on Kiss1 neurons for the fast adaptation to changing energy levels. Nonetheless, the main regulatory centers of leptin action must lie upstream of Kiss1$^{ARH}$ neurons because, as mentioned previously, the removal of Lepr from Kiss1 neurons does not lead to an evident reproductive phenotype in mice (*Cravo et al., 2013*; *Donato et al., 2011a*)—although specific metabolic challenges have not been performed in mice lacking Lepr from Kiss1 neurons. Moreover, Kiss1$^{ARH}$ neurons are largely glutamatergic (*Nestor et al., 2016*) while the main action of leptin to regulate reproduction (and metabolism) occurs through GABAergic neurons (*Martin et al., 2014*; *Vong et al., 2011*; *Zuure et al., 2013*). Therefore, the reproductive phenotype observed in male *Kiss1$^{Cre}$:Nhlh2$^{fl/fl}$* mice suggests that in addition to the action of leptin, Nhlh2 must relay additional information from other unknown factor/s that add a strong regulatory component in the activation of *Kiss1* and *Tac2* genes in males.

Overall, this study identified a TF (Nhlh2) that highly marks KNDy neurons in the adult mouse brain with strong enhancement capability of the *KISS1* and *TAC3* promoter activity. The action of Nhlh2 appears to partially explain the reproductive (but not metabolic) phenotype observed in whole-body Nhlh2 KO mice and constitutes a novel metabolic pathway for the central regulation of reproductive function.

## Materials and methods

### Mice

*Kiss1-Nhlh2* conditional knockout mice were generated by crossing *Nhlh2$^{fl/fl}$* and Kiss1$^{Cre:GFP}$ knock-in mice. *Nhlh2$^{fl/fl}$* mice (RRID:MGI:5524018) were obtained from Dr. Thomas Braun (Max Planck

Institute, Germany) (*Schmid et al., 2013*) and Kiss1^Cre:GFP (RRID:MGI:6278139) were obtained from Dr. Richard Palmiter (University of Washington, Seattle, WA) (*Padilla et al., 2018*). Animals were maintained under constant conditions of temperature (22–24°C) and light (12 hr light [06:00]/dark [18:00] cycle), fed with standard mouse chow (Teklad F6 Rodent Diet 8664) and were given ad libitum access to tap water. *Kiss1^Cre:Nhlh2^fl/fl* males or females between age 8 and 20 weeks were used and studied in parallel to control *Nhlh2^fl/fl* and Kiss1^Cre/+ het littermates. Genotyping was conducted by PCR analyses on isolated genomic DNA from tail biopsies.

## Reagents

The agonist of NK3R (senktide) was purchased from Tocris Biosciences (Minneapolis, MN); mouse kisspeptin-10 (Kp-10) and recombinant mouse leptin were obtained from Phoenix Pharmaceuticals (Burlingame, CA). All drugs were dissolved in saline (0.9% NaCl). Doses and timings for hormonal analyses were selected on the basis of previous studies (*Navarro et al., 2012a*; *Navarro et al., 2015*; *León et al., 2016*).

## Experimental design

### General procedures

For intracerebroventricular (icv) injection, 2–3 days before the experiment, the mice were briefly anesthetized with isoflurane and a small hole was bored in the skull 1 mm lateral and 0.5 mm posterior to bregma with a Hamilton syringe attached to a 27-gauge needle fitted with polyethylene tubing, leaving 3.5 mm of the needle tip exposed. Once the initial hole was made, all subsequent injections were made at the same site. For icv injections, mice were anesthetized with isoflurane for a total of 2–3 min, during which time 5 µl of solution were slowly and continuously injected into the lateral ventricle. The needle remained inserted for approximately 60 s after the injection to minimize backflow up the needle track. Mice typically recovered from the anesthesia within 3 min after the injection. For hormonal analyses, blood samples (4 µl) were obtained from the tail and stored at −80°C until hormonal determination. The dose and time of collection were selected based on our previous studies (*Navarro et al., 2015*).

## Identification of highly enriched *Nhlh2* transcripts in Kiss1 neurons

### Drop-Seq analysis

The enrichment of *Nhlh2* in arcuate Kiss1 cells was determined using the previously published Drop-Seq database (*Campbell et al., 2017*). Briefly, the experiments used a total of 53 adults (4–12 weeks old) virgin male and female mice and processed in five sample batches. Libraries were sequenced on the Illumina NextSeq500. Read 1 was 20 bp (bases 1–12 cell barcode, bases 13–20 UMI), read 2 (paired-end) was 60 bp, and the index primer was 8 bp (on multiplexed samples). 20,921 transcriptomes from acutely dissociated Arc-ME cells of adult mice were profiled. After correcting for batch effects, principal component (PC) analysis was performed. About 25 PCs were chosen for further all-cell clustering analyses and used as input for t-distributed stochastic neighbor embedding, implemented by the Seurat software package with the perplexity parameter set to the default, 30 (*Macosko et al., 2015*; *Satija et al., 2015*). The tSNE procedure returns a two-dimensional embedding of single cells, with cells that have similar expression signatures of genes within the variable set located near each other in the embedding. To identify cell types, a density clustering approach was implemented. Clusters with fewer than 10 cells and those containing expression markers for more than one canonical cell type (e.g., neuron, oligodendrocyte, and tanycyte), representing cell doublets (two cells in a single droplet), were removed. Kiss1 neurons were determined by the enrichment of *Kiss1*, *Tac2*, *Tacr3*, and *Pdyn*.

### RNAscope In situ hybridization

To (1) evaluate the co-expression of *Kiss1* and *Nhlh2* mRNA in key reproductive brain nuclei (ARH and AVPV), and (2) validate the *Kiss1^Cre:Nhlh2^fl/fl* mouse model, dual fluorescence ISH (RNAscope) was performed using the Multiplex Fluorescent Detection Kit v2 as recommended (Advanced Cell Diagnostic, 323110), in tissue samples from intact male mice (n=2), using probes against *Nhlh2* (527811-C2) and *Kiss1* (500141-C1). The brains were removed for ISH, fresh frozen on dry ice, and then stored at −80°C until sectioned. Five sets of 20 µm sections in the coronal plane were cut on a

cryostat, from the diagonal band of Broca to the mammillary bodies, thaw mounted onto SuperFrost Plus slides (VWR Scientific) and stored at −80°C. A single set was used for ISH experiment (adjacent sections 100 μm apart). For quantification of Kiss1 neurons expressing Nhlh2, images were taken at 20× magnification of the sections containing AVPV, PeN, and the three rostro-to-caudal levels of the ARH, and Kiss1 neurons expressing Nhlh2 were identified using ImageJ, counted, and subtracted from the total number of kiss1 neurons.

## Determination of Nhlh2 binding sites

The identification of Nhlh2 binding sites on human, mouse, and rat *Kiss1* and *Tac2/3* genes was performed using JASPAR (*Fornes et al., 2020*). In short, 1500 bp of each gene regulatory region (−1000 bp to +500 bp, TSS=+1) were extracted from Esembl and used to search NHLH2 sites with two available matrices (PF0015.1 and MA1529.1). A relative score cutoff of 0.8 was used to identify putative sites, as represented in *Figure 2*. Transcript used for this analysis are: human *KISS1* ENST00000367194.5, mouse *Kiss1* ENSMUST00000007433.4, rat *Kiss1* ENSRNOT00000077054.1, human *TAC3* ENST00000300108.7, mouse *Tac2* ENSMUST00000026466.4, and rat *Tac3* ENSR-NOT00000005679.2. Positional output of Jaspar analysis can be found in the *Supplementary file 1*.

## Functional promoter assays

To determine whether NHLH2 alters the transcriptional activity of putative target genes (*KISS1* and *TAC3*) we transfected Neuro2A cells (N2A, ATCC, Manassas, VA) with luciferase reporter constructs (*Mueller et al., 2011*) containing the 5′ flanking region of these genes, in addition to an NHLH2 expression vector (OHu05087-Genscript). The cells (ATCC, Manassas, VA) were cultured in a humidified atmosphere containing 5% $CO_2$ and 37°C. They were maintained in DMEM containing high glucose (4.5 g $l^{-1}$; Sigma-Aldrich), supplemented with 10% fetal bovine serum (FBS) (Invitrogen), Glutamine (Sigma-Aldrich), 100 U $ml^{-1}$ penicillin, and 100 μg $ml^{-1}$ streptomycin (Invitrogen). For the assays, the cells (400,000 cells per well) were seeded onto 24-well plates in DMEM containing 10% FBS. After 24 h, the reporter constructs (both in the luciferase reporter plasmid pGL2) were transiently co-transfected along with NHLH2 for 5 hr using Lipofectamine 2000 (Invitrogen) at a ratio (1 μg DNA:2.5 μl Lipofectamine 2000) in Optimem (Invitrogen). After transfection, the cells were returned to serum containing DMEM medium; 24 hr later, they were harvested and assayed for luciferase activity using the Firefly Luciferase Glow Assay Kit (Pierce, Rockford, IL). The assay was performed in opaque 96-well plates and light emission was measured in a Spectramax M5 microplate reader (Molecular Devices, Sunnyvale, CA). Transfection efficiency was normalized by co-transfecting the plasmid CMV-Sport-β-gal (Invitrogen) at 10 ng $ml^{-1}$ and determining β-Galactosidase activity using the Tropix Galacto Light Plus (ABI) as reported previously (*Heger et al., 2007*). Experiments were performed three times.

## Chromatin immunoprecipitation assays and PCR detection of chromatin immunoprecipitated DNA

ChIP assays were performed in Rat-1 cells (Thermo Fisher Scientific, Waltham, MA) transfected with the NHLH2 expression vector. The cells were cultured at 37°C in a humidified atmosphere containing 5% $CO_2$ and maintained in DMEM containing high glucose (4.5 g $l^{-1}$; Sigma-Aldrich), supplemented with 10% FBS (Invitrogen), Glutamine (Sigma-Aldrich), 100 U $ml^{-1}$ penicillin, and 100 μg $ml^{-1}$ streptomycin (Invitrogen). In short, 2 million cells were seeded onto 10 cm diameter dishes in DMEM containing 10% FBS. After 24 h, cells were transiently transfected with NHLH2-FLAG or empty vector for 5 hr using Lipofectamine 2000 (Invitrogen) at a ratio (1 μg DNA:2.5 μl Lipofectamine 2000) in Optimem (Invitrogen). After transfection, the cells were returned to serum containing DMEM medium and, 48 hr later, they were snap frozen for subsequent chromatin extraction and ChIP assay. The ChIP procedure was carried out essentially as previously described using 3 μg antibody against the FLAG/OctA epitope (Santa Cruz Biotechnology, sc-807, Dallas, TX) (*Lomniczi et al., 2013*; *Lomniczi et al., 2015*; *Toro et al., 2018*; *Vazquez et al., 2018*). Cells were washed once in ice-cold phosphate-buffered saline (PBS) containing a protease inhibitor cocktail (PI, 1 mM phenylmethylsulfonylfluoride, 7 μg $ml^{-1}$ aprotinin, 0.7 μg $ml^{-1}$ pepstatin A, and 0.5 μg $ml^{-1}$ leupeptin), a phosphatase inhibitor cocktail (PhI, 1 mM β-glycerophosphate, 1 mM sodium pyrophosphate, and 1 mM sodium fluoride), and an HDAC inhibitor (20 mM sodium butyrate). Thereafter, cells were cross-

linked by exposing them to 1% formaldehyde for 10 min at room temperature. After two additional washing steps in PBS the samples were lysed with 200 µl SDS buffer (0.5% SDS, 50 mM Tris-HCl, and 10 mM EDTA) containing protease, phosphatase, and HDAC inhibitors and sonicated for 45 s to yield chromatin fragments of approximately 500 base pairs (bp) using the microtip of a Fisher Scientific FB 705 sonicator. Size fragmentation was confirmed by agarose gel electrophoresis. The sonicated chromatin was clarified by centrifugation at 14,000 rpm for 10 min at 4°C, brought up to 1 ml in Chip Dilution Buffer (CDB) (16.7 mM Tris-HCl, pH 8.1, 150 mM NaCl, 1.2 mM EDTA, 1.1% Triton X-100, and 0.01% SDS) containing the PI and PhI cocktails, and the HDAC inhibitor described above. The samples were then stored at −80°C for subsequent immunoprecipitation. For this step, chromatin was pre-cleared with Protein G beads (Dynabeads, Invitrogen, Carlsbad, CA) for 1 hr at 4°C. About 25 µl aliquots of chromatin were then incubated with 3 µg of anti-FLAG antibody. Antibody-chromatin complexes and 25 µl of protein G beads solution (Dynabeads) were incubated at 4°C overnight with gentle agitation. Immunocomplexes were washed sequentially with 0.5 ml low salt wash buffer (20 mM Tris-HCl, pH 8.1, 150 mM NaCl, 2 mM EDTA, 1% Triton X-100%, and 0.1% SDS), high salt wash buffer (20 mM Tris-HCl, pH 8.1, 500 mM NaCl, 2 mM EDTA, 1% Triton X-100%, and 0.1% SDS), LiCl buffer (10 mM Tris-HCl, pH 8.1, 250 M LiCl, 1% Nonidet P-40, 1% sodium deoxycholate, and 1 mM EDTA), and with TE buffer (10 mM Tris-HCl, pH 8.0, and 1 mM EDTA). The immunocomplexes were eluted with 100 µl of 0.1 M NaHCO$_3$% and 1% SDS at 65°C for 45 min. Cross-linking was reversed by adding 4 µl of 5 M NaCl and incubating at 95°C for 30 min. DNA was recovered by using ChIP DNA Clean and Concentrator columns (Zymo Research, Irvine, CA) and stored at −80°C before PCR analysis. All chemicals were purchased from Sigma-Aldrich (St. Louis, MO).

For PCR detection of chromatin immunoprecipitated DNA, the promoter regions of the genes of interest were amplified by PCR. The primers (Eurofins MWG Operon, Huntsville, Al) used to amplify the 5′ flanking region of the genes of interest are listed below. PCR reactions were performed using 1 µl of each immunoprecipitate (IP) or input samples, primer mix (1 µM each primer) and Pfx DNA Polymerase (Thermo Fisher Scientific, Waltham, MA) in a final volume of 25 µl. Input samples consisted of 10% of the chromatin volume used for immunoprecipitation. The thermocycling conditions used were 95°C for 5 min, followed by 36 cycles of 15 s at 95°C, 30 s at 57°C, and 60 s at 68°C. PCR products were run in a 1.5% agarose gel and stained with ethidium bromide. The primers used are listed in *Table 1*.

## Reproductive maturation of *Kiss1$^{Cre}$:Nhlh2$^{fl/fl}$* male and female mice

In order to assess the reproductive phenotype of mice lacking Nhlh2 on Kiss1 neurons, we generated a *Kiss1$^{Cre}$:Nhlh2$^{fl/fl}$* mouse as indicated above. Prepubertal littermate *Nhlh2$^{fl/fl}$* (n=19) and *Kiss1-Cre:Nhlh2$^{fl/fl}$* (n=9) males were monitored daily from postnatal day 26 for preputial separation as an indirect marker of puberty onset. BW was measured at the day of puberty onset.

In females, littermate *Nhlh2$^{fl/fl}$* (n=15) and *Kiss1$^{Cre}$:Nhlh2$^{fl/fl}$* (n=10) were monitored daily from postnatal 26 for BW and pubertal progression (VO as indicated by complete canalization of the vagina) and littermate *Nhlh2$^{fl/fl}$* (n=14) and *Kiss1$^{Cre}$:Nhlh2$^{fl/fl}$* (n=9) mice were subsequently monitored for first estrus (first day with cornified cells determined by daily morning vaginal cytology) for 15 days after the day of VO. In addition, estrous cyclicity was monitored by daily vaginal cytology, for a period of 30 days, in 3-month-old *Nhlh2$^{fl/fl}$* (n=8) and *Kiss1$^{Cre}$:Nhlh2$^{fl/fl}$* (n=9). Cytology samples were obtained every morning (10 a.m.) and placed on a glass slide for determination of the estrous cycle stage under the microscope as previously described (*Martin et al., 2014*).

## Characterization of the estradiol-induced luteinizing hormone surge

In this experiment, *Nhlh2$^{fl/fl}$* (n=6) and *Kiss1$^{Cre}$:Nhlh2$^{fl/fl}$* (n=6) adult female mice were subjected to bilateral ovariectomy (OVX) via abdominal incision under light isoflurane anesthesia. Immediately after OVX, capsules filled with E$_2$ (1 µg/20 g BW) were implanted subcutaneously (sc) via a small midscapular incision at the base of the neck; 5 days later, mice were subcutaneously injected in the morning with estradiol benzoate (1 µg/20 g BW) to produce elevated proestrus-like E$_2$ levels (LH surge) on the following day. Blood samples were collected at 10:00 hr and 19:30 hr (*Czieselsky et al., 2016*); LH levels were measured via ELISA. Additionally, to assess co-expression of *Kiss1* and Nhlh2 mRNA in the AVPV (key area for LH surge onset), dual fluorescence ISH was

**Table 1.** Primers used for ChIP.

| Gene | Accession # | Primers | Sequence | Amplicon (bp) |
|------|-------------|---------|----------|---------------|
| *Kiss1* | NM_181692.1 | rKiss1p CHIP-F | TCGGGCAGCCAGATAGAGGAAGC | 91 |
| | | rKiss1p CHIP-R | TTGAGGGCCGAGGGAGAAGAG | |
| *Kiss1* | NM_181692.1 | rKiss1e CHIP-F | CCAGCCCGGGGGATGGGGTGTAA | 192 |
| | | rKiss1e CHIP-R | AGGGGCCAGCGCAGCAGCACAAT | |
| *Tac3* | NM_019162.2 | rTAC2p CHIP-F | ACGTGCGTGTCTGGGTATGTGA | 123 |
| | | rTAC2p CHIP-R | GGAGGGTTTGGGGGAGTCG | |
| | | rPCSK1p CHIP-R | CCTTCGAGACAGCATTCACA | |

F, forward; R, reverse.

performed in additional tissue samples from OVX+E$_2$ control and *Kiss1$^{Cre}$:Nhlh2$^{fl/fl}$* mice (n=2) as we described previously.

## Fecundity test in *Kiss1$^{Cre}$:Nhlh2$^{fl/fl}$* male and female mice

In this experiment, adult control (n=4) or *Kiss1$^{Cre}$:Nhlh2$^{fl/fl}$* (n=4–7) littermate male mice (>75 days) were placed with proven fertile WT females for 3 months and time to delivery and number of pups per litter were monitored. In females, the fertility assessment was performed by breeding adult WT (n=4) or *Kiss1$^{Cre}$:Nhlh2$^{fl/fl}$* (n=4) females with WT males previously proven to be father litters. The time to first litter and number of pups per litter were monitored.

Additionally, the histology of the testes was analyzed in a separate cohort of adult (3-month-old) Nhlh2$^{fl/fl}$ and *Kiss1$^{Cre}$:Nhlh2$^{fl/fl}$* mice (n=3/group). Testes were collected and fixed in Bouin's solution. The tissues were embedded in paraffin and sectioned (10 μm) for hematoxylin and eosin staining (Harvard Medical School Rodent Pathology Core) and images acquired under 10× magnification. In females, the ovarian ultra-structure was also analyzed in adult (3-month-old) Nhlh2 $^{fl/fl}$ and *Kiss1$^{Cre}$: Nhlh2$^{fl/fl}$* mice (n=3/group). Ovaries were collected and processed as described above for the testes. The images acquired under 5× magnification and the ovaries were analyzed for the presence of CLs per section. Each value represents the number of CLs of 1 representative section from the middle line of one ovary per animal.

## Expression of *Kiss1*, *Tac2*, *Pdyn*, and *Tacr3* in the ARH of male mice

We aimed to determine whether there are changes in the expression of *Kiss1*, *Tac2*, *Pdyn*, and *Tacr3* in the MBH, the site that includes the ARH, between adult controls, Kiss1$^{Cre/+}$(n=4) or *Kiss1$^{Cre}$: Nhlh2$^{fl/fl}$* (n=4) male and female mice. The hypothalamus was dissected taking as limits the posterior margin of the optic chiasm (rostrally) and the anterior margin of the mammillary bodies (caudally), with a dissection depth of approximately 2 mm. Each hypothalamic sample was dissected and divided into two, the suprachiasmatic region (the preoptic area, POA) or the MBH, and fragments were stored at −80℃ until further processing. Total RNA from the MBH was isolated using TRIzol reagent (Invitrogen) followed by chloroform/isopropanol extraction. RNA was quantified using a NanoDrop 2000 spectrophotometer (Thermo Fisher Scientific), and 1 μm of RNA was reverse transcribed using an iScript cDNA synthesis kit (Bio-Rad). Semi-quantitative real-time PCR assays were performed on an ABI Prism 7000 sequence detection system and analyzed using ABI Prism 7000 SDS software (Applied Biosystems). The cycling conditions were the following: 2 min incubation at 95℃ (hot start), 45 amplification cycles (95℃ for 30 s, 60℃ for 30 s, and 45 s at 75℃, with fluorescence detection at the end of each cycle), followed by a melting curve of the amplified products obtained by a ramped increase in temperature from 55℃ to 95℃ to confirm the presence of a single amplification product per reaction. For data analysis, we used the 2(-Delta Delta C(T)) method and target gene was standardized to Hprt levels in each sample. The primers used are listed in *Table 2*.

In addition, *Kiss1* mRNA expression was analyzed by in situ hybridization in brain slices. Images were taken with a Leica DM2500 microscope and processed using ImageJ (NIH). The content of Kiss1 was presented as the percentage of the specific brain area (ARH) occupied by Kiss1 cells. Brain area was delimited using as reference the mouse coronal section from the Allen Brain Atlas (mouse. brain-map.org/static/atlas).

**Table 2.** Primers used for real-time RT-PCR.

| Gene | Accession # | Primers | Sequence |
|------|-------------|---------|----------|
| *Hprt* | NM_013556.2 | Hprt-F | CCTGCTGGATTACATTAAAGCGCTG |
| | | Hprt-R | GTCAAGGGCATATCCAACAACAAAC |
| *Kiss1* | AF472576.1 | Kiss1-F | GCTGCTGCTTCTCCTCTGTG |
| | | Kiss1-R | TCTGCATACCGCGATTCCTT |
| *Tac1* | NM_009311.2 | Tac1-F | ATGAAAATCCTCGTGGCCGT |
| | | Tac1-R | GTTCTGCATCGCGCTTCTTT |
| *Pdyn* | NM_018863.4 | Pdyn-F | ACAGGGGGAGACTCTCATCT |
| | | Pdyn-R | GGGGATGAATGACCTGCTTACT |
| *Tacr3* | NM_021382.6 | Tacr3-F | GCCATTGCAGTGGACAGGTAT |
| | | Tacr3-R | ACGGCCTGGCATGACTTTTA |
| *Nhlh2* | NM_178777.3 | Nhlh2-F | ACCAGAAGAGCCAAGAAGCCA |
| | | Nhlh2-R | GCGGGTGTATGGTTGTTCACTTAG |

F, forward; R, reverse.

## Expression of *Nhlh2* in the gonads

We analyzed the expression of *Nhlh2* in the testes and ovaries of Kiss1$^{Cre/+}$ controls and *Kiss1*$^{Cre}$:*Nhlh2*$^{fl/fl}$ mice. We followed the protocol previously described. The primers used are listed in *Table 2*.

## Gene expression of *Kiss1* and *Nhlh2* in the ARH during the prepubertal period

We aimed to determine changes in the expression of *Kiss1* and *Nhlh2* in the MBH of prepubertal WT male and female mice. To this end, we analyzed the expression profile of *Kiss1* and *Nhlh2* in the MBH at ages infantile (p10), early juvenile (p15), late juvenile (p20–22), and prepubertal (p25–30) by RT-qPCR, normalized by the housekeeping gene *Hprt* (n=5–6/group). We followed the protocol previously described. The primers used are listed in *Table 2*.

## Effect of Kp-10 and senktide administration on LH release in *Kiss1*$^{Cre}$:*Nhlh2*$^{fl/fl}$ male mice

Hormonal (LH) responses to known stimulators of GnRH and/or gonadotropin secretion were studied in *Nhlh2*$^{fl/fl}$ and *Kiss1*$^{Cre}$:*Nhlh2*$^{fl/fl}$ male mice. The mice were injected with mouse kisspeptin-10 (Kp-10) (50 pmol/mouse in 5 µl/icv); Nhlh2$^{fl/fl}$ (n=10) and *Kiss1*$^{Cre}$:*Nhlh2*$^{fl/fl}$ (n=7) or Senktide (600 pmol/mouse in 5 µl/icv); Nhlh2$^{fl/fl}$ (n=9) and *Kiss1*$^{Cre}$:*Nhlh2*$^{fl/fl}$ (n=7), and blood samples were obtained at 30 min after icv administration. Doses and routes of administration were selected in the basis of previous references (*García-Galiano et al., 2012a*; *García-Galiano et al., 2012b*).

## Characterization of the postgonadectomy response of LH in *Kiss1*$^{Cre}$:*Nhlh2*$^{fl/fl}$ males

Bilateral removal of testes from 3- to 4-month-old males was performed with light isoflurane anesthesia. Briefly, the ventral skin was shaved and cleaned to perform one small incision in the skin and abdominal musculature of the abdomen. Once the gonads were identified and excised, the muscle incision was sutured, and the skin was closed with surgical clips. LH levels were measured in *Nhlh2*$^{fl/fl}$ (n=10) and *Kiss1*$^{Cre}$:*Nhlh2*$^{fl/fl}$ (n=7) male mice. Blood samples were collected before and 10 days after bilateral gonadectomy (GDX).

## Characterization of the LH pattern in response of fasting/refeeding in *Kiss1*$^{Cre}$:*Nhlh2*$^{fl/fl}$ males

The hormonal (LH) response to fasting and refeeding in gonadectomized *Nhlh2*$^{fl/fl}$ (n=10) and *Kiss1*$^{Cre}$:*Nhlh2*$^{fl/fl}$ (n=7) male mice was determined. Food was removed at 8:00 a.m. and replaced 24 hr

later. Blood samples were obtained before and 6 hr, 12 hr, and 24 hr after the initiation of fasting, as well as 6 hr and 12 hr after initiation of refeeding. Additionally, we determined the LH response to leptin in *Nhlh2*$^{fl/fl}$ (n=9) and *Kiss1*$^{Cre}$:*Nhlh2*$^{fl/fl}$ (n=6) male mice after overnight fasting. The mice were injected with recombinant mouse Leptin (2 μg/mouse in 5 μl/icv), and blood samples were obtained at 30 min after icv administration (*Ross et al., 2018*).

## Hormone measurements

LH was measured by a sensitive sandwich ELISA for assessment of whole blood LH concentrations as previously described elsewhere (*Steyn et al., 2013*). A 96-well high-affinity binding microplate (9018; Corning) was coated with 50 μl of capture antibody (monoclonal antibody, anti-bovine LH β subunit, 518B7; RRID:AB_2665514, University of California) at a final dilution of 1:1000 (in 1× PBS, 1.09 g of Na$_2$HPO$_4$ [anhydrous], 0.32 g of NaH$_2$PO$_4$ [anhydrous], and 9 g of NaCl in 1000 ml of distilled water) and incubated overnight at 4°C. To minimize unspecific binding of the capture antibody, wells were incubated with 200 μl of blocking buffer (5% [weight/volume] skim milk powder in 1× PBS-T [1× PBS with 0.05% Tween 20]) for 2 hr at room temperature. A standard curve was generated using a twofold serial dilution of LH (reference preparation, AFP- 5306A; National Institute of Diabetes and Digestive and Kidney Diseases National Hormone and Pituitary Program [NIDDK-NHPP]) in 0.2% (weight/volume) BSA-1× PBS-T. The LH standards and blood samples were incubated with 50 μl of detection antibody (polyclonal antibody, rabbit LH antiserum, and AF-P240580Rb; RRID:AB_2665533, NIDDK-NHPP) at a final dilution of 1:10,000 for 1.5 hr (at room temperature). Each well containing bound substrate was incubated with 50 μl of horseradish peroxidase-conjugated antibody (1:2000; Cat# 1706515; RRID:AB_2617112; Bio-Rad, Hercules, CA). After a 1.5 hr incubation, 100 Ul of o-phenylenediamine (002003; Invitrogen) substrate containing 0.1% H$_2$O$_2$ was added to each well and left at room temperature for 30 min. The reaction was stopped by the addition of 50 μl of 3 M HCl to each well, and the absorbance of each well was read at a wavelength of 490 nm (Sunrise; Tecan Group). The concentration of LH in each blood sample was determined by comparing the optical density (OD) values of the experimental LH samples to the OD values of the known LH standard curve. The reported intra- and inter-assay coefficients of variation for this assay are 6.05% and 4.29%, respectively (*Steyn et al., 2013*). The functional sensitivity of the ELISA assay was 0.0039 ng/ml with a CV% of 3.3%.

## Statistical analysis

The data are expressed as median±max/min values (represented by violin plots) or ± SEM for each group. In the violin plots, the minimum and maximum values and the distribution of the data are presented. A two-tailed unpaired t-Student test or a one- or two-way ANOVA test followed by Tukey or Bonferroni or Fisher's post hoc test was used to assess variation among experimental groups. Significance level was set at p<0.05. All analyses were performed with GraphPad Prism Software, Inc (San Diego, CA).

## Study approval

All animal care and experimental procedures were approved by the National Institute of Health, and Brigham and Women's Hospital Institutional Animal Care and Use Committee, protocol #05165. The Brigham and Women's Hospital is a registered research facility with the U.S. Department of Agriculture (#14–19), is accredited by the American Association for the Accreditation of Laboratory Animal Care and meets the National Institutes of Health standards as set forth in the Guide for the Care and Use of Laboratory Animals (DHHS Publication No. (NIH) 85-23 Revised 1985).

## Acknowledgements

The authors thank Dr. Thomas Braun (Max Planck Institute, Germany) for providing the Nhlh2$^{fl/fl}$ mouse, and Dr. John N Campbell (Department of Biology, University of Virginia, Charlottesville, VA) for his contribution in the analysis of the DropSeq data. This work was supported by Grants R01HD090151, R01HD099084, R21HD095383 by the Eunice Kennedy Shriver National Institute of Child Health and Human Development (NICHD) and National Institute of Health (NIH) to VMN, F32HD097963 by the NIH to EAM, 1R01HD084542 to AL, 8P51OD011092 for the operation of the

Oregon National Primate Research Center, R37HD019938, R01 HD082314, and the BWH Women's Brain Initiative to UBK.

## Additional information

### Funding

| Funder | Grant reference number | Author |
|---|---|---|
| Eunice Kennedy Shriver National Institute of Child Health and Human Development | R01HD090151 | Víctor M Navarro |
| Eunice Kennedy Shriver National Institute of Child Health and Human Development | R01HD099084 | Víctor M Navarro |
| Eunice Kennedy Shriver National Institute of Child Health and Human Development | R21HD095383 | Víctor M Navarro |
| Eunice Kennedy Shriver National Institute of Child Health and Human Development | F32HD097963 | Elizabeth A McCarthy |
| Eunice Kennedy Shriver National Institute of Child Health and Human Development | R01HD084542 | Alejandro Lomniczi |
| Eunice Kennedy Shriver National Institute of Child Health and Human Development | R37HD019938 | Ursula B Kaiser |
| Eunice Kennedy Shriver National Institute of Child Health and Human Development | R01 HD082314 | Ursula B Kaiser |

The funders had no role in study design, data collection and interpretation, or the decision to submit the work for publication.

### Author contributions

Silvia Leon, Data curation, Formal analysis, Validation, Investigation, Visualization, Methodology, Writing - original draft, Writing - review and editing; Rajae Talbi, Data curation, Formal analysis, Validation, Investigation, Methodology, Writing - original draft, Writing - review and editing; Elizabeth A McCarthy, Chrysanthi Fergani, Data curation, Investigation, Methodology; Kaitlin Ferrari, Investigation, Methodology; Lydie Naule, Ursula B Kaiser, Resources; Ji Hae Choi, Investigation; Rona S Carroll, Supervision, Investigation; Carlos F Aylwin, Resources, Validation, Investigation; Alejandro Lomniczi, Conceptualization, Resources, Software, Formal analysis, Validation, Investigation, Methodology; Víctor M Navarro, Conceptualization, Resources, Data curation, Software, Formal analysis, Supervision, Funding acquisition, Validation, Investigation, Visualization, Methodology, Writing - original draft, Project administration, Writing - review and editing

### Author ORCIDs

Rajae Talbi ![ORCID] https://orcid.org/0000-0001-7158-6246
Chrysanthi Fergani ![ORCID] http://orcid.org/0000-0001-7028-4158
Ursula B Kaiser ![ORCID] http://orcid.org/0000-0002-8237-0704
Víctor M Navarro ![ORCID] https://orcid.org/0000-0002-5799-219X

### Ethics

Animal experimentation: All animal care and experimental procedures were approved by the National Institute of Health, and Brigham and Women's Hospital Institutional Animal Care and Use Committee, protocol #05165. The Brigham and Women's Hospital is a registered research facility with the U.S. Department of Agriculture (#14-19), is accredited by the American Association for the Accreditation of Laboratory Animal Care and meets the National Institutes of Health standards as

set forth in the Guide for the Care and Use of Laboratory Animals (DHHS Publication No. (NIH) 85-23 Revised 1985).

## Decision letter and Author response

Decision letter https://doi.org/10.7554/eLife.69765.sa1
Author response https://doi.org/10.7554/eLife.69765.sa2

## Additional files

### Supplementary files

• Supplementary file 1. Jaspar Analysis of Nhlh2 binding sites.

• Transparent reporting form

### Data availability

All data generated or analysed during this study are included in the manuscript and supporting files.

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
