## [Decision Letter]

**Acceptance summary:**

The identification of a specific helix-loop-helix transcription factor that regulates the synthesis of kisspeptin, a vital peptide for controlling gonadotropin hormone releasing hormone, greatly expands our understanding of the neural control of reproduction. Ablation of the transcription factor specifically in kisspeptin neurons of the arcuate nucleus delayed puberty in males while leaving females relatively unperturbed, highlighting the numerous ways in which fundamental mechanisms, even those central to reproduction, can differ between the sexes.

**Decision letter after peer review:**

Thank you for submitting your article "Sex specific pubertal and metabolic regulation of Kiss1 neurons via Nhlh2" for consideration by *eLife*. Your article has been reviewed by 3 peer reviewers, including Margaret M McCarthy as the Reviewing Editor and Reviewer #1, and the evaluation has been overseen by Catherine Dulac as the Senior Editor.

Essential revisions:

In addition to the specific comments to the authors below, the reviewers agree that:

1) Confirmation of specificity to the arcuate needs to include evidence that expression of Nhlh2 in the periphery, particularly the gonads, is not impacting puberty.

2) Data on the reduction in Nhlh2 expression presented in Figure 2 for females should be expanded to males.

*Reviewer #1:*

This manuscript began with analysis of a published dataset from drop-seq analysis of the arcuate nucleus and medien eminence and identified a transcription factor, Nhlh2, enriched in kisspeptin neurons of the arcuate. The authors reasoned it may be important to kisspeptin transcription and took off from there.

The strengths of the manuscript are the systematic demonstration of the transcriptional power of Nhlh2 over kisspeptin, and neurokinB, the assessment of the impact of deletion of Nhlh2 from only KNDy neurons in the arcuate on pubertal onset, LH surges (in females) and fertility in both sexes and the integration of responses to metabolic changes. The authors conclude that Nhlh2 regulates the response of Kiss1 neurons to leptin, closing the circle on how the timing of puberty, without damaging fertility, is achieved. The manuscript is impeccably written with appropriate caveats and limitations noted.

The authors refer to PN25 as "peripubertal" (line 152), what is this based on? It seems clearly "prepubertal" to me.

Figure 3 – Why only females? Does the same phenomenon happen in males?

Supplemental Figure 3 seems to be missing.

*Reviewer #2:*

Leon et al., investigated the role of Nhlh2 in Kiss1 cells in regulating puberty, adult fertility, and metabolism. Using a previously published dataset on gene expression in the arcuate nucleus, the authors identified the transcription factor Nhlh2 as a potential candidate for mediating the metabolic effects on reproduction. The authors find that Nhlh2 expression decreases prior to puberty, at least in females, and is co-expressed in ~72% of arcuate kisspeptin cells in adult males. A transgenic mouse was created to knockout Nhlh2 in kisspeptin cells (Kiss1Nhlh2 mice) and reproductive and metabolic outcomes were assessed. Puberty onset was delayed in male, but not female, Kiss1Nhlh2 mice. Adult female Kiss1Nhlh2 mice had longer estrous cycles, but were otherwise fertile. Adult male Kiss1Nhlh2 mice had a varying degree of fertility deficits, as well as a 50% reduction in Kiss1 and 25% reduction in Tacr3 expression in the arcuate. The administration of Kp-10 or senktide, a Tacr3 agonist, produced normal responses in LH release, indicating there are no deficits downstream of ARC kisspeptin neurons in Kiss1Nhlh2 mice. Lastly, male Kiss1Nhlh2 mice had a more severe decrease in LH levels in response to metabolic fasting, but Kiss1Nhlh2 mice and control had a comparable increase in LH following feeding. Leptin treatment resulted in an increase in LH in control and Kiss1Nhlh2 mice, but to a lesser extent in Kiss1Nhlh2 mice. These data indicate that in males, Nhlh2 may play a role in puberty onset and fertility, as well as a role in mediating the effects of leptin on arcuate kisspeptin neurons.

Strengths: Nhlh2 is identified as a novel candidate regulating kisspeptin neurons and Kiss1 expression. The role of Nhlh2 in kisspeptin cells on both puberty onset and adult fertility were examined in both sexes, with metabolism effects studied in males. The differential effects of Nhlh2 deletion in kisspeptin cells suggest that Nhlh2 is a novel prime candidate in understanding how the regulation of reproduction and metabolism differ between male and females.

Weaknesses: The normal expression levels of Nhlh2 prior to puberty and in adulthood are only presented in one sex, prepubertal females and adult males. Thus, it is difficult to understand how the loss of Nhlh2 in Kiss1 cells at a certain age might impact the outcomes. It also makes it difficult to fully interpret the sex differences in reproductive outcomes. In addition, it is unclear if Nhlh2 is expressed in other parts of the body where Kiss1 is also expressed, such as the gonads, and thus, the deletion of Nhlh2 may not be specific to Kiss1 cells in the arcuate nucleus.

The manuscript by Leon et al., examines the role of Nhlh2 in Kiss1 cells in puberty onset, fertility, and metabolic regulation. The data show that a knockdown of Nhlh2 in Kiss1 cells altered male puberty and fertility (in some males), while females had fewer, less severe deficits. Nhlh2 in Kiss1 cells does not appear to play a role in metabolic regulation of Kiss1 cells in males. The manuscript is well written, the experiments are interesting, and the results indicate a new candidate for moderating male puberty onset. The results and the conclusions could be strengthened by providing expression data for both males and females prepubertally and in adulthood. Here are a few comments for the authors to consider:

1. Figure 1 demonstrates the co-expression of Kiss1 and Nhlh2 in males, but were similar studies performed in females? Is it possible the less severe effects in female Kiss1/Nhlh2 KOs are due to less co-expression of Nhlh2 and Kiss1 in the ARC of females? Knowing the degree of co-expression in both sexes would be important to understanding and interpreting any effects in females and the potential sex differences found.

2. Is Nhlh2 present in ARC Kiss1 neurons prior to puberty in males? Figure 3 provides this data for females, but not males. It would be helpful to also see this qPCR data in males to understand how Nhlh2 levels prior to development (and any sex differences) relate to the puberty and fertility data and the sex differences in the effects.

3. The authors suggest there is a possible downregulation of Nhlh2 expression due to sex steroid levels in prepubertal mice (Figure 3 female qPCR data). Is it known if Nhlh2 expression is regulated by sex steroids in prepubertal, pubertal, and/or adult mice? Considering that Kiss1 expression in the hypothalamus is regulated by sex steroids, at least in part, is it likely Nhlh2 is also regulated by sex steroids? A brief discussion on these points and how they relate to the current data would be beneficial to the reader.

4. Is Nhlh2 co-expressed in other parts of the body that also express Kiss1, like the testes or ovaries? Data from humans indicates there is minimal expression of Nhlh2 in the testes. Is this similar in mice and how might that impact the data and the sex differences found? It would be helpful to address if the knockout was specific to ARC Kiss1 cells.

5. Males showed an approximately 50% reduction in Kiss1 expression in the ARC (Figure 6). How does the degree of reduction in Kiss1 expression vary between males and does that relate to the somewhat bimodal fertility phenotype in males? It is unclear if this reduction is Kiss1 is potentially a cause of male fertility issues in some males or if all males have this reduction and thus, the fertility deficits are likely due to something else.

6. There seem to be a few cases where one data point appears to be driving an effect (e.g. fertility data and Figure 7E). Is there any concern that one individual is driving significant effects here or have the authors considered a nonparametric test here?

7. Relatedly, the statistical analysis section states all data are presented as means and range/SEM, but there are violin plots presenting the median. A description of these plots should be added to the analysis section.

8. Line 137-142: refers to figure 3, but it seems it should be Figure 2.

9. In Figure 5, the figure legends state that (B) is the number of litters in 3 months and (C) is total number of pups, but that is reversed in the actual Figure.

10. In Figure 5 legend, it states groups with different letters are significantly different, but there are no letters in this Figure, presumably because there are no statistical differences. Can the authors please clarify?

*Reviewer #3:*

Based on previous evidence that Nhlh2-/- mice present disrupted pubertal transition and fertility, and that Nhlh2 expression is enriched in the arcuate hypothalamus, this study evaluates whether Nhlh2 action within kisspeptin neurons of the reproductive axis are responsible for the phenotype. The authors demonstrate the NHLH2 can influence of expression of both Kiss1 and Tac2/3. Conditional removal of Nhlh2 from Kiss1-expressing cells did not phenocopy the global knockout, but did induce delayed puberty, in males. Furthermore, the authors show that Kiss1 Nhlh2 KO mice are more sensitive to metabolic gating of LH, compared to controls. However, because Kiss1 is also expressed in the gonads, it is unclear if the reported phenotypes are central, as argued by the authors. The manuscript lacks detail of the functional and molecular phenotype of the gonads in either males or females, which is important to substantiate the claims.

In figure 3 the authors "hypothesize that the overall expression of Nhlh2 throughout development would decrease prior to puberty onset as it becomes restricted to Kiss1 neurons." I believe to test this hypothesis you would need to do the anatomy to see if Nhlh2 transcript is more broadly expressed early and then becomes restricted. Instead, the authors only assess Nhlh2 expression from a punch of ARH tissue. Based on this evidence alone, it is possible that the gene is downregulated over postnatal development, as opposed to more "restricted."

Figure 4: Kiss1 is also expressed in the periphery, and in particular in the gonadal organs. Does NHLH2 regulate Kiss1 expression in the testes or ovaries?

Figures4 and 5. Males show delayed puberty and mover variable fertility. Were the gonads and T assessed? This seems important to the story and should be compared to the global KO.

Figure 6. It seems important to also include females here. Is female gene expression profile similar to males? Was the data in Figure 2 specific to males?

Figure 7. The data here seem to support that the observed phenotype may not be central.

Fig7d. The KOs are more sensitive to metabolic gating. Was this expected, why? The finding seems inconsistent with the gene expression profile presented in Figure 2.

Figure 1B, middle panel: is this just the red channel alone? The staining looks yellow in the densest areas, as if green is present in the image. Can you comment on this? Also, based on the expression profile in A (from Campbell et al.,), wouldn't you expect to see Nhlh2 expression in POMC neurons. In B, a region of the ARH that should also contain POMC neurons, Nhlh2 expression appears exclusive to Kiss1 neurons. Also, how was the 71% co-expression determined, is the data not shown? Aside from the ISH stain, the rest of this figure has been previously published. If you exclude part B, the figure should be reported as supplementary info.

Lines 137-142: I believe this should reference Figure 2, not Figure 3.

Figure 7e. Are basal levels, fasted?

---

## [Author Response]

Essential revisions:In addition to the specific comments to the authors below, the reviewers agree that:1) Confirmation of specificity to the arcuate needs to include evidence that expression of Nhlh2 in the periphery, particularly the gonads, is not impacting puberty.

We appreciate this very important concern. New analysis of Nhlh2 expression in the gonads of male and female Kiss1-Nhlh2KO and controls clearly show no difference in Nhlh2 transcript (new supplemental figure 2), which supports the central effect described in the study.

2) Data on the reduction in Nhlh2 expression presented in Figure 2 for females should be expanded to males.

Thank you for this suggestion. An ontogeny for males has been added to the revised figure 3. The expression of Kiss1 and Nhlh2 in males follow that observed in females.

Reviewer #1:This manuscript began with analysis of a published dataset from drop-seq analysis of the arcuate nucleus and medien eminence and identified a transcription factor, Nhlh2, enriched in kisspeptin neurons of the arcuate. The authors reasoned it may be important to kisspeptin transcription and took off from there.The strengths of the manuscript are the systematic demonstration of the transcriptional power of Nhlh2 over kisspeptin, and neurokinB, the assessment of the impact of deletion of Nhlh2 from only KNDy neurons in the arcuate on pubertal onset, LH surges (in females) and fertility in both sexes and the integration of responses to metabolic changes. The authors conclude that Nhlh2 regulates the response of Kiss1 neurons to leptin, closing the circle on how the timing of puberty, without damaging fertility, is achieved. The manuscript is impeccably written with appropriate caveats and limitations noted.

Thank you for the overall positive assessment of the study.

The authors refer to PN25 as "peripubertal" (line 152), what is this based on? It seems clearly "prepubertal" to me.

This reviewer is correct, and it has been corrected in the edited manuscript.

Figure 3 – Why only females? Does the same phenomenon happen in males?

An ontogeny for males has been added to the revised figure 3. The expression of Kiss1 and Nhlh2 in males follow that observed in females.

Supplemental Figure 3 seems to be missing.

We thank you for detecting this oversight. All supplemental figures have been included in the revised manuscript.

Reviewer #2:Leon et al., investigated the role of Nhlh2 in Kiss1 cells in regulating puberty, adult fertility, and metabolism. Using a previously published dataset on gene expression in the arcuate nucleus, the authors identified the transcription factor Nhlh2 as a potential candidate for mediating the metabolic effects on reproduction. The authors find that Nhlh2 expression decreases prior to puberty, at least in females, and is co-expressed in ~72% of arcuate kisspeptin cells in adult males. A transgenic mouse was created to knockout Nhlh2 in kisspeptin cells (Kiss1Nhlh2 mice) and reproductive and metabolic outcomes were assessed. Puberty onset was delayed in male, but not female, Kiss1Nhlh2 mice. Adult female Kiss1Nhlh2 mice had longer estrous cycles, but were otherwise fertile. Adult male Kiss1Nhlh2 mice had a varying degree of fertility deficits, as well as a 50% reduction in Kiss1 and 25% reduction in Tacr3 expression in the arcuate. The administration of Kp-10 or senktide, a Tacr3 agonist, produced normal responses in LH release, indicating there are no deficits downstream of ARC kisspeptin neurons in Kiss1Nhlh2 mice. Lastly, male Kiss1Nhlh2 mice had a more severe decrease in LH levels in response to metabolic fasting, but Kiss1Nhlh2 mice and control had a comparable increase in LH following feeding. Leptin treatment resulted in an increase in LH in control and Kiss1Nhlh2 mice, but to a lesser extent in Kiss1Nhlh2 mice. These data indicate that in males, Nhlh2 may play a role in puberty onset and fertility, as well as a role in mediating the effects of leptin on arcuate kisspeptin neurons.Strengths: Nhlh2 is identified as a novel candidate regulating kisspeptin neurons and Kiss1 expression. The role of Nhlh2 in kisspeptin cells on both puberty onset and adult fertility were examined in both sexes, with metabolism effects studied in males. The differential effects of Nhlh2 deletion in kisspeptin cells suggest that Nhlh2 is a novel prime candidate in understanding how the regulation of reproduction and metabolism differ between male and females.Weaknesses: The normal expression levels of Nhlh2 prior to puberty and in adulthood are only presented in one sex, prepubertal females and adult males. Thus, it is difficult to understand how the loss of Nhlh2 in Kiss1 cells at a certain age might impact the outcomes. It also makes it difficult to fully interpret the sex differences in reproductive outcomes. In addition, it is unclear if Nhlh2 is expressed in other parts of the body where Kiss1 is also expressed, such as the gonads, and thus, the deletion of Nhlh2 may not be specific to Kiss1 cells in the arcuate nucleus.

Thank you for the overall positive assessment of the study and identification of weaknesses, which have been addressed as indicated below.

The manuscript by Leon et al., examines the role of Nhlh2 in Kiss1 cells in puberty onset, fertility, and metabolic regulation. The data show that a knockdown of Nhlh2 in Kiss1 cells altered male puberty and fertility (in some males), while females had fewer, less severe deficits. Nhlh2 in Kiss1 cells does not appear to play a role in metabolic regulation of Kiss1 cells in males. The manuscript is well written, the experiments are interesting, and the results indicate a new candidate for moderating male puberty onset. The results and the conclusions could be strengthened by providing expression data for both males and females prepubertally and in adulthood. Here are a few comments for the authors to consider:1. Figure 1 demonstrates the co-expression of Kiss1 and Nhlh2 in males, but were similar studies performed in females? Is it possible the less severe effects in female Kiss1/Nhlh2 KOs are due to less co-expression of Nhlh2 and Kiss1 in the ARC of females? Knowing the degree of co-expression in both sexes would be important to understanding and interpreting any effects in females and the potential sex differences found.

This is an excellent point that we have addressed in the revised manuscript. The % of co-expression in females is lower than in males (39% vs 71%), which could account for the less severe reproductive phenotype in females.

2. Is Nhlh2 present in ARC Kiss1 neurons prior to puberty in males? Figure 3 provides this data for females, but not males. It would be helpful to also see this qPCR data in males to understand how Nhlh2 levels prior to development (and any sex differences) relate to the puberty and fertility data and the sex differences in the effects.

An ontogeny for males has been added to the revised figure 3. The expression of Kiss1 and Nhlh2 follow that observed in females.

3. The authors suggest there is a possible downregulation of Nhlh2 expression due to sex steroid levels in prepubertal mice (Figure 3 female qPCR data). Is it known if Nhlh2 expression is regulated by sex steroids in prepubertal, pubertal, and/or adult mice? Considering that Kiss1 expression in the hypothalamus is regulated by sex steroids, at least in part, is it likely Nhlh2 is also regulated by sex steroids? A brief discussion on these points and how they relate to the current data would be beneficial to the reader.

Thank you for bringing up the possible regulation of Nhlh2 by sex steroids. Unfortunately, we do not have direct evidence of Nhlh2 expression comparing GDX vs GDX+T/E2 mice. Nonetheless, we concur that it is likely that Nhlh2 is regulated by sex steroids as part of the machinery that regulates KNDy neurons during the negative feedback of sex steroids. This has been added to the discussion (lines 265-267).

4. Is Nhlh2 co-expressed in other parts of the body that also express Kiss1, like the testes or ovaries? Data from humans indicates there is minimal expression of Nhlh2 in the testes. Is this similar in mice and how might that impact the data and the sex differences found? It would be helpful to address if the knockout was specific to ARC Kiss1 cells.

We appreciate this very important concern. New analysis of Nhlh2 expression in the gonads of male and female Kiss1-Nhlh2KO and controls clearly show no difference in Nhlh2 transcript (new supplemental figure 2), which supports the central effect described in the study.

5. Males showed an approximately 50% reduction in Kiss1 expression in the ARC (Figure 6). How does the degree of reduction in Kiss1 expression vary between males and does that relate to the somewhat bimodal fertility phenotype in males? It is unclear if this reduction is Kiss1 is potentially a cause of male fertility issues in some males or if all males have this reduction and thus, the fertility deficits are likely due to something else.

This reviewer’s rationale is a possibility. However, the fertility was performed on different males of which we unfortunately could not collect brains from, thus a direct comparison of fertility and Kiss1 expression cannot be made in these specific experiments.

6. There seem to be a few cases where one data point appears to be driving an effect (e.g. fertility data and Figure 7E). Is there any concern that one individual is driving significant effects here or have the authors considered a nonparametric test here?

These data follow a Gaussian distribution and, therefore, we believe they must be assessed using ANOVA. However, to assess whether a single value in figure 7 was driving the significance, we removed it and performed the same statistics. We still obtained a significant difference. Because we do not have any reason to consider that value as an outlier (it is within 2 SD from the mean), we kept it in the figure as it was.

7. Relatedly, the statistical analysis section states all data are presented as means and range/SEM, but there are violin plots presenting the median. A description of these plots should be added to the analysis section.

This error has been corrected in the revised manuscript.

8. Line 137-142: refers to figure 3, but it seems it should be Figure 2.

This error has been corrected in the revised manuscript.

9. In Figure 5, the figure legends state that (B) is the number of litters in 3 months and (C) is total number of pups, but that is reversed in the actual Figure.

This error has been corrected in the revised manuscript.

10. In Figure 5 legend, it states groups with different letters are significantly different, but there are no letters in this Figure, presumably because there are no statistical differences. Can the authors please clarify?

As the reviewer points out, the reason for the absence of letters is because there are no statistical differences. This error has been corrected in the revised manuscript.

Reviewer #3:Based on previous evidence that Nhlh2-/- mice present disrupted pubertal transition and fertility, and that Nhlh2 expression is enriched in the arcuate hypothalamus, this study evaluates whether Nhlh2 action within kisspeptin neurons of the reproductive axis are responsible for the phenotype. The authors demonstrate the NHLH2 can influence of expression of both Kiss1 and Tac2/3. Conditional removal of Nhlh2 from Kiss1-expressing cells did not phenocopy the global knockout, but did induce delayed puberty, in males. Furthermore, the authors show that Kiss1 Nhlh2 KO mice are more sensitive to metabolic gating of LH, compared to controls. However, because Kiss1 is also expressed in the gonads, it is unclear if the reported phenotypes are central, as argued by the authors. The manuscript lacks detail of the functional and molecular phenotype of the gonads in either males or females, which is important to substantiate the claims.In figure 3 the authors "hypothesize that the overall expression of Nhlh2 throughout development would decrease prior to puberty onset as it becomes restricted to Kiss1 neurons." I believe to test this hypothesis you would need to do the anatomy to see if Nhlh2 transcript is more broadly expressed early and then becomes restricted. Instead, the authors only assess Nhlh2 expression from a punch of ARH tissue. Based on this evidence alone, it is possible that the gene is downregulated over postnatal development, as opposed to more "restricted."

Thank you for this assessment and valid concern. We have revised the discussion of this data in the manuscript.

Figure 4: Kiss1 is also expressed in the periphery, and in particular in the gonadal organs. Does NHLH2 regulate Kiss1 expression in the testes or ovaries?

New analysis of Nhlh2 expression in the gonads of male and female Kiss1-Nhlh2KO and controls clearly show no difference in Nhlh2 transcript (new supplemental figure 2), which supports the central effect described in the study.

Figures 4 and 5. Males show delayed puberty and mover variable fertility. Were the gonads and T assessed? This seems important to the story and should be compared to the global KO.

Unfortunately, we didn´t measure T levels in these animals, however, the normal basal levels of LH, and in response to kisspeptin, in addition to the normal testicular histology and mature sperm (new supplemental figure 4) support an overall normal function of the HPG axis in most adult males.

Figure 6. It seems important to also include females here. Is female gene expression profile similar to males? Was the data in Figure 2 specific to males?

We appreciate this concern and, as indicated above to the other reviewers, Figures 3 and 6 have been revised to include both sexes. Thus, we have found that both sexes show the same expression profile of Nhlh2 postnatally (new figure 3) and during adulthood, Kiss1-Nhlh2KO mice of both sexes show decreased Kiss1 levels compared to controls.

Figure 7. The data here seem to support that the observed phenotype may not be central.

We respectfully disagree. First the gonads of these KO mice are normal (i.e. normal signs of gametogenesis) and express the same amount of Nhlh2 as controls. Second, it has been shown that just 5% of Kiss1 expression is sufficient in males to maintain reproductive competence. Thus, it is likely that even with a >50% decrease in kisspeptin expression in adulthood in KO mice, reproduction is maintained. This indicates that puberty onset, which is delayed in KO males, might be more sensitive to metabolic cues and the action of Nhlh2.

Fig7d. The KOs are more sensitive to metabolic gating. Was this expected, why? The finding seems inconsistent with the gene expression profile presented in Figure 2.

It is expected that in the absence of a factor that relays the stimulatory action of a metabolic factor, e.g. leptin, in its absence, the expression of Kiss1 and Tac3 be decreased faster as stimulatory factors such as leptin decrease. This is the same reason why mice respond with a diminished LH release after exogenous leptin treatment (Figure 7).

Figure 1B, middle panel: is this just the red channel alone? The staining looks yellow in the densest areas, as if green is present in the image. Can you comment on this?

The quality of the images has been improved to prevent that areas of high intensity staining appeared yellow before merging them due to saturation.

Also, based on the expression profile in A (from Campbell et al.), wouldn't you expect to see Nhlh2 expression in POMC neurons. In B, a region of the ARH that should also contain POMC neurons, Nhlh2 expression appears exclusive to Kiss1 neurons.

This is an excellent observation and as the reviewer can appreciate in the tSNE plot, the expression of Nhlh2 in POMC neurons is less than that in Kiss1 neurons. Therefore, it is likely that the amount of Nhlh2 transcript in POMC neurons is not sufficient to be detected in all of the animals. However, in supplemental figure 2A, where we show the validation of the Kiss1-Nhlh2KO model, non Kiss1 neurons that express Nhlh2 are detected, which are likely POMC neurons based on their location.

Also, how was the 71% co-expression determined, is the data not shown?

This description has been expanded in the manuscript.

Aside from the ISH stain, the rest of this figure has been previously published. If you exclude part B, the figure should be reported as supplementary info.

The tSNE plot analysis of was assessed specifically for Nhlh2, which was not included in the original manuscript (Campbell et al). However, because the raw data used was indeed previously published, we have moved that figure to the supplements.

Lines 137-142: I believe this should reference Figure 2, not Figure 3.

This error has been corrected in the revised manuscript.

Figure 7e. Are basal levels, fasted?

Yes, basal LH levels were collected after overnight fasting as indicated in the methods.